# A noninvasive fluorescence imaging-based platform measures 3D anisotropic extracellular diffusion

Peng Chen [1,6], Xun Chen[1,6], R. Glenn Hepfer[1,2], Brooke J. Damon[1], Changcheng Shi[1,3], Jenny J. Yao[4], Matthew C. Coombs[1,2], Michael J. Kern[5], Tong Ye [1,5 ✉] & Hai Yao [1,2 ✉]

Diffusion is a major molecular transport mechanism in biological systems. Quantifying direction-dependent (i.e., anisotropic) diffusion is vitally important to depicting how the three-dimensional (3D) tissue structure and composition affect the biochemical environment, and thus define tissue functions. However, a tool for noninvasively measuring the 3D anisotropic extracellular diffusion of biorelevant molecules is not yet available. Here, we present light-sheet imaging-based Fourier transform fluorescence recovery after photobleaching (LiFT-FRAP), which noninvasively determines 3D diffusion tensors of various biomolecules with diffusivities up to 51 $\mu m^2\ s^{-1}$, reaching the physiological diffusivity range in most biological systems. Using cornea as an example, LiFT-FRAP reveals fundamental limitations of current invasive two-dimensional diffusion measurements, which have drawn controversial conclusions on extracellular diffusion in healthy and clinically treated tissues. Moreover, LiFT-FRAP demonstrates that tissue structural or compositional changes caused by diseases or scaffold fabrication yield direction-dependent diffusion changes. These results demonstrate LiFT-FRAP as a powerful platform technology for studying disease mechanisms, advancing clinical outcomes, and improving tissue engineering.

---

[1] Clemson-MUSC Joint Bioengineering Program, Department of Bioengineering, Clemson University, Clemson, SC, USA. [2] Department of Oral Health Sciences, Medical University of South Carolina, Charleston, SC, USA. [3] Ningbo Institute of Industrial Technology, Chinese Academy of Sciences, Ningbo, Zhejiang, China. [4] Department of Physics, Harvard University, Cambridge, MA, USA. [5] Department of Regenerative Medicine and Cell Biology, Medical University of South Carolina, Charleston, SC, USA. [6] These authors contributed equally: Peng Chen, Xun Chen. ✉email: ye7@clemson.edu; haiyao@clemson.edu

Extracellular diffusion study is one of the main focuses of current biotransport research. As a major molecular transport mechanism in tissues, extracellular diffusion is critical to tissue functions[1–7]. To maintain cellular homeostasis and growth, essential nutrients and signaling molecules must be diffused through the highly organized interstitial tissue space. For instance, in avascular tissues such as intervertebral discs, extracellular diffusion distances of essential nutrients (e.g., glucose) through the well-organized disc lamellae structures can exceed 10 mm- much greater than that for intracellular diffusion[6]. The impedance of extracellular nutrient diffusion in these discs causes cell death, leading to disc degeneration-associated low back pain, a major cause of disability worldwide[6]. Extracellular diffusion is also crucial to tissue patterning during development, as the action range of signaling molecules is determined by their extracellular diffusion properties[2,4,8]. Besides maintaining normal tissue function, diffusion also plays a key role in tissue engineering and drug delivery, and must be well-tuned by the extracellular environment to ensure interventional success[1,3,9].

Tissue-level diffusion of biomolecules is dependent on both tissue architecture and solute properties[10]. Given the three-dimensional (3D) organization of biological tissues and direction-dependent (i.e., anisotropic) alignment of matrix structures (Fig. 1a), extracellular diffusion is spatially complex. Thus, the extracellular diffusion of a specific solute is best represented not by a single diffusivity value, but rather a 3D diffusion tensor[11] (Fig. 1b). As evidenced by computational modeling, 3D anisotropic diffusion generates significantly different molecular concentration distributions compared to direction-independent (i.e., isotropic) diffusion[12–14] (Fig. 1c and Supplementary Note 1). These altered concentration profiles and diffusion rates yield significant downstream effects on cellular responses and tissue function[12–14]. However, the precise 3D extracellular diffusion behavior of most biomolecules remains largely unknown.

Currently, biotransport research field in studying molecular diffusion is suffering from a lack of analytical tools capable of quantitative 3D diffusion measurement of various biomolecules in biological tissues, as well as biomaterials[15]. Such hurdle has significantly impacted biomedical basic science research for understanding pathophysiology, translational research for developing clinical treatment, and biotech industry research and development for standardizing the biofabrication process. The primary existing methods for diffusion measurement, including diffusion cells[16,17], electroconductivity method[18], and optical fluorescence techniques[15], are either one-dimensional (1D) or two-dimensional (2D), being unable to directly quantify the 3D anisotropic extracellular diffusion of biomolecules. Therefore, in most diffusion studies, the third-dimension diffusion is ignored or assumed to be the same as the other two dimensions[19–22], resulting in the misrepresentation of actual molecular dynamics in biological tissues or biomaterials. Recently, increasing efforts have been made to determine the 3D diffusion properties in various biological tissues (e.g., intervertebral discs[17,18,21], temporomandibular joint discs[23], and corneas[24]) based on the existing 1D or 2D approaches through physical tissue sectioning. However, such a strategy suffers from invasiveness and labor-intensiveness. Invasive tissue dissection procedures damage tissue integrity and release physiological mechanical stress stored in the extracellular matrix network, drastically altering tissue structure and thereby extracellular biomolecule diffusion[23,25]. Such is the case for extracellular diffusion measurement in cornea tissues, which has gained unresolved controversy due to conflicting results. 3D diffusion measurements in excised cornea tissues using 2D fluorescence correlation spectroscopy exhibited isotropic extracellular diffusion[24]. Conversely, in situ magnetic resonance imaging (MRI)-based 3D diffusion tensor imaging in intact ovine corneas reported anisotropic extracellular diffusion[26]. Since changes in diffusion behavior caused by the invasive procedures are unquantifiable, it is difficult to tell the actual 3D diffusion property in tissues. Consequently, the inability to accurately quantify the baseline 3D diffusion properties in biological tissues prevents the understanding of how diseases or treatments affect the 3D diffusion properties, as well as the design of biomimetic scaffolds resembling tissue 3D diffusion properties. Therefore, there is a critical unmet need to develop a noninvasive 3D diffusion measurement tool to advance biotransport research.

While two noninvasive 3D methods are currently available-MRI-based diffusion tensor imaging[27] (broadly used in brain imaging) and diffusion tensor optical coherence tomography[11], their use is limited to only water molecules or large scattering nanoparticles, respectively. Thus, to accurately assess 3D extracellular diffusion of a range of biorelevant molecules, we must seek new tools for noninvasive 3D measurement. In biotransport research, fluorescence recovery after photobleaching (FRAP) is one of the most powerful techniques that allow for diffusion measurement of various molecules[15]. In a FRAP experiment, fluorescent molecules within a small region are irreversibly photobleached by a laser pulse. The surrounding fluorescent molecules subsequently diffuse into the photobleached region, resulting in fluorescence recovery. Then, a theoretical model extracts the molecular diffusivity from the fluorescence recovery imaging data[28]. A highly versatile tool, FRAP has been widely used to measure intra- and extracellular 2D diffusion of various molecules in all kinds of biological systems[15,29]. FRAP methods have since been developed to extend diffusion measurements to 3D samples, including two-photon and confocal laser scanning microscopy-based FRAP techniques[30–34]. However, these 3D approaches assume isotropic diffusion and are unable to generate a 3D anisotropic diffusion tensor. Other methods, such as line FRAP[10], elliptical surface photobleaching[35], fluorescence imaging of continuous point photobleaching[36], and 2D Fourier transform FRAP[20,37,38], have been developed to investigate 2D anisotropic diffusion in tissues. However, they are unable to measure diffusion along the optical axis. To attempt to overcome this limitation, 2D measurements from three orthogonal tissue sections from the same sample have been combined to reconstruct the 3D diffusion tensor[21,23]. However, as previously discussed, such an approach is invasive and risks altering the tissue diffusion behavior. Noninvasively measuring 3D anisotropic diffusion remains the long-standing goal of the FRAP technique in the biotransport research field[15]. To address the consequences of tissue dissection, recently, we successfully developed a 3D FRAP technique that can directly extract 3D diffusion tensors from 3D recordings of fluorescence recovery images from a single FRAP experiment. This method enabled noninvasive 3D extracellular diffusion measurement in biological tissues[25]. However, there remain significant limitations for detecting the physiological diffusion rates of most biomolecules. The slow 3D volumetric imaging speed of conventional spot-scanned two-photon microscopy limits the diffusion measurements for solute diffusivities at $0.5\ \mu m^2\ s^{-1}$ and lower[25], rendering it impossible to investigate meaningful 3D molecular diffusion in biological systems (Supplementary Table 1).

Here, we present a 3D diffusion measurement platform: light-sheet imaging-based Fourier transform FRAP (LiFT-FRAP), which integrates a FRAP data acquisition system and our 3D FRAP spatial Fourier transform data analysis method[25], to achieve noninvasive 3D diffusion tensor measurements in the physiological diffusivity range—up to $51\ \mu m^2\ s^{-1}$—of most biomolecules in biological tissues. Using LiFT-FRAP, we resolve a long-standing question in cornea physiology, which is whether the molecular diffusion in cornea tissues is isotropic or

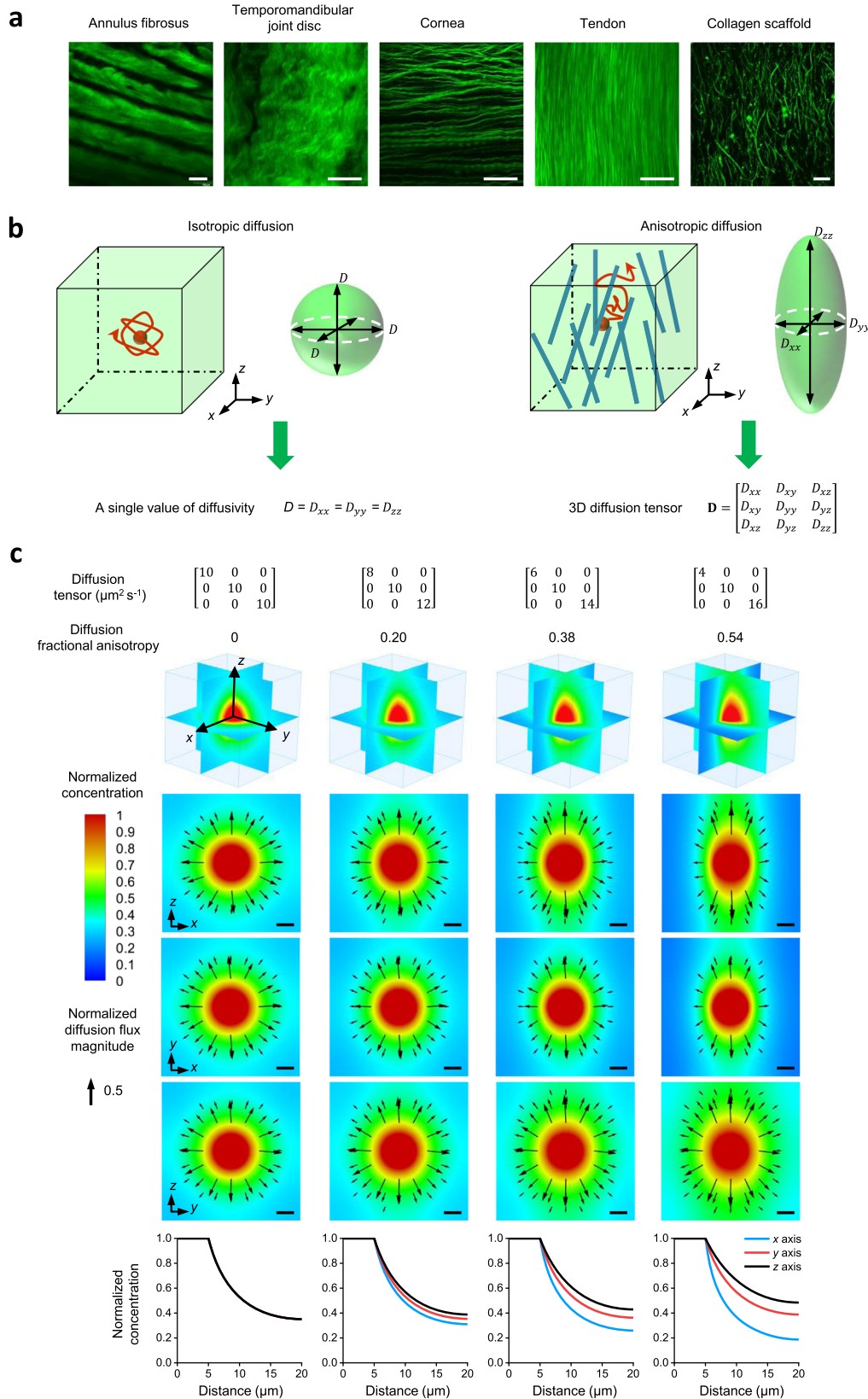

anisotropic. We further quantify how a clinical treatment changes the 3D cornea diffusion behaviors that are misrepresented with 2D measurements. Our results demonstrate that the 3D diffusion of fluorescent molecules was indeed highly anisotropic in native corneas under physiological conditions. Critically, compared to the native cornea, our noninvasive 3D measurements revealed

that collagen crosslinking (CXL) treatment, a recent Food and Drug Administration (FDA)-approved clinical intervention for keratoconus[39], actually reduces 3D diffusion anisotropy, countering the enhanced diffusion anisotropy observed in 2D diffusion measurement with excised CXL corneas. These findings demonstrate the distortion of extracellular diffusion caused by invasive

**Fig. 1 3D diffusion tensor is required to characterize anisotropic molecular diffusion in biological systems. a** Anisotropic matrix structure imaged by second-harmonic generation imaging in different biological tissues: rat annulus fibrosus, porcine temporomandibular joint disc, porcine cornea, rat tail tendon, and collagen scaffold. Experiments were repeated in at least 3 independent samples. Scale bar, 50 μm. **b** 3D isotropic diffusion with identical diffusivities in all directions can be characterized by a single diffusivity value. 3D anisotropic diffusion in biological systems with direction-dependent diffusivities is best described by a 3D diffusion tensor. Diffusion is considered isotropic when the diagonal components of the 3D diffusion tensor are not significantly different, and anisotropic when they are significantly different. **c** Computational simulation experiments of molecules diffusing out from a source center in a 3D domain demonstrate that molecular concentration distributions are regulated by 3D molecular diffusion (Supplementary Note 1). 3D anisotropic diffusion generates significantly different molecular distributions compared to isotropic diffusion. Top panel: 3D diffusion tensors with an identical average diffusivity (10 μm$^2$ s$^{-1}$) used in computational modeling and calculated diffusion fractional anisotropies used to show the extent of diffusion anisotropy. Middle panel: 3D normalized concentration distributions and 2D normalized concentration distributions and diffusion flux vectors (black arrows) in three orthogonal planes shown in the 3D plot at a time of 28 s. The concentrations were normalized to the value at the source center while the diffusion fluxes were normalized to the maximum diffusion flux magnitude among all models. Scale bar, 5 μm. Bottom panel: Normalized concentration plots along x axis (blue), y axis (red), and z axis (black) from the source center to the edge of the 3D domain at a time of 28 s. Under 3D anisotropic diffusion, concentration profiles vary among the axes, and the extent of this difference becomes greater as diffusion anisotropy increases. The simulation is for illustration purposes and is not related to the actual FRAP experiments. Source Data is available as a Source Data file for **c**.

2D diffusion measurement and confirm the necessity of non-invasive 3D diffusion measurement for accurately assessing the biological consequences of clinical interventions on solute transport and tissue homeostasis. Moreover, we apply LiFT-FRAP to study the 3D diffusion changes in response to diseases in tendon tissues. Finally, the application of LiFT-FRAP on biomimetic scaffolds demonstrates that LiFT-FRAP enables delineating the impact of fiber alignment and solute size on 3D diffusion behavior in tissue engineering bioscaffolds, therefore providing insights on scaffold design and offering a valuable tool for the standardizing of biofabrication process. Overall, our results quantitatively show that tissue structural or compositional changes caused by clinical treatment, diseases, or scaffold fabrication yield direction-dependent diffusion changes. Through these measurements, we demonstrate the broad utility of LiFT-FRAP as a powerful platform technique to noninvasively investigate physiological 3D extracellular diffusion of biomolecules in biological systems, allowing for the advancement of clinical treatments, study of disease mechanisms, and characterization and functional optimization of tissue engineering scaffolds.

## Results

**LiFT-FRAP for noninvasive fast 3D diffusion tensor measurement.** Noninvasive 3D diffusion tensor measurement of fast-diffusing molecules was achieved by the development of a LiFT-FRAP data acquisition system and an accompanying 3D FRAP data analysis method. The LiFT-FRAP system, which integrates a two-photon 3D volume bleaching generator for confined 3D bleaching with a high-speed two-photon scanned light-sheet microscope, allowed fast volumetric image acquisition of the 3D fluorescence recovery process (Fig. 2a and Supplementary Fig. 1). Within the LiFT-FRAP system, a light sheet was generated and rapidly scanned to illuminate a 3D volume, a high-speed camera was used to capture the fluorescence signals, and an independent bleaching path was implemented to bleach a small 3D region at the center of illuminated 3D volume. The scanning apparatus and piezo stage in the LiFT-FRAP system were well-controlled to assure coordinated operation and high-quality 3D imaging (Supplementary Figs. 2, 3 and Supplementary Notes 2, 3). For each LiFT-FRAP experiment, both 3D prebleaching images and postbleaching images were recorded. Then our 3D spatial Fourier transform method, which includes frequency domain data conversion, curve fitting, and tensor calculation, was applied to analyze the 3D image data and extract the 3D diffusion tensor (**D**), a 3 × 3 symmetric matrix ($D_{ij} = D_{ji}$)[40] through a single FRAP experiment (Fig. 2b and Supplementary Fig. 4, Supplementary Notes 4, 5).

Compared to the volumetric imaging rate of 0.05 volumes per second in conventional spot-scanned two-photon microscopy[25], LiFT-FRAP system achieved an imaging rate of 8 volumes per second (160 times faster), enabling capture of rapid 3D fluorescence recovery with a time resolution of 125 ms (Fig. 3a and Supplementary Movie 1). To validate the performance of LiFT-FRAP, we determined the 3D diffusion tensors of standard fluorescent probes in homogeneous solutions and compared the molecular diffusivity results with 2D FRAP measurements and theoretical predictions. Diffusion in homogeneous solutions is considered isotropic[25,35]; therefore, the diffusivities attained from 2D and 3D experiments are directly comparable. According to the Stokes–Einstein equation[41], molecular diffusivity in homogenous solutions is determined by both solution viscosity and molecular size. First, we determined the 3D diffusion tensors of sodium fluorescein (376 Da) in glycerol solutions with different viscosities (glycerol concentrations 55%, 60%, 70%, and 80%) (Supplementary Fig. 5a and Supplementary Movie 2). As anticipated, the 3D diffusion of sodium fluorescein was isotropic in all four solutions, and diffusivity decreased as solution viscosity increased (Fig. 3b). With our current setup, the maximum diffusivity measured using LiFT-FRAP was 51 μm$^2$ s$^{-1}$, 100-fold higher than our previously reported value of 0.5 μm$^2$ s$^{-1}$, demonstrating the capacity of LiFT-FRAP to quantify rapid 3D diffusion for a variety of biomolecules (Supplementary Table 1). Notably, LiFT-FRAP yielded comparable diffusion results to both the 2D FRAP measurements as well as theoretical predictions (Fig. 3c and Supplementary Table 2, Supplementary Note 6). Next, we measured the 3D diffusion tensors of fluorescein isothiocyanate-conjugated dextran (FD) molecules with different molecular weights (4, 10 and 40 kDa) (FD4, FD10, and FD40) in 60% glycerol solution (Supplementary Fig. 5b and Supplementary Movie 3). As expected, the 3D diffusion for all FD molecules was isotropic, and diffusivity decreased as molecular size (i.e., molecular weight) increased (Fig. 3d). The FD molecule diffusivities measured by LiFT-FRAP also agreed well with 2D FRAP values and theoretical predictions (Fig. 3e and Supplementary Table 3). These results demonstrate that LiFT-FRAP robustly achieves fast, accurate 3D diffusion tensor measurement.

**LiFT-FRAP reveals misrepresentation of extracellular diffusion in cornea tissues by invasive 2D measurement.** Having demonstrated that LiFT-FRAP is a robust tool for fast 3D diffusion tensor measurement, we then applied this noninvasive LiFT-FRAP technique to identify 3D diffusion properties in cornea tissues (Fig. 4a and Supplementary Fig. 6a). We also performed 2D FRAP measurements in exercised cornea sections (Fig. 4b) to reveal how the invasive dissection procedures altered

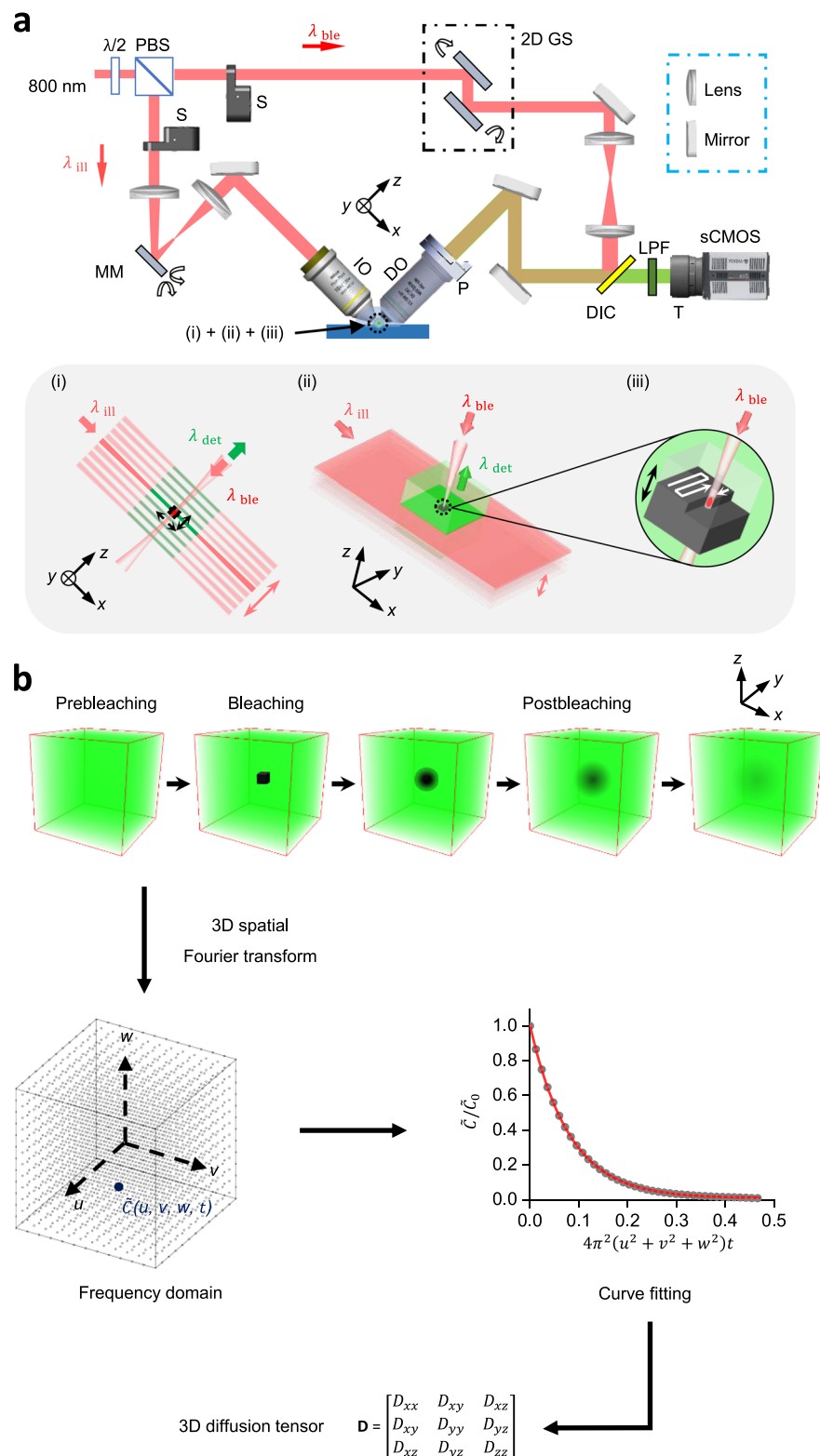

the diffusion measurement results, therefore demonstrating the necessity of the noninvasive LiFT-FRAP technique. Since cornea tissues are subjected to tensile stress exerted by the intraocular pressure under physiological conditions, the invasive dissection procedure involved in the 2D FRAP study will unwittingly release the mechanical stress, therefore resulting in tissue structure changes. Second-harmonic generation (SHG) imaging of the collagen structure in intact eyeballs and exercised cornea sections

confirmed such structure changes caused by tissue dissection: collagen fibers were wavier and more crimped in exercised cornea sections than in intact eyeballs (Fig. 4c and Supplementary Fig. 6b). Morphology analysis with the SHG images showed that the structure coherency coefficient, a morphometric indicator of structural anisotropy, was lower in dissected cornea sections than in intact eyeballs (Fig. 4d). Moreover, we found no significant difference in structure coherency coefficients in intact eyeballs

**Fig. 2 LiFT-FRAP for noninvasive fast 3D diffusion tensor measurement. a** Schematic of the LiFT-FRAP system. A light sheet illuminates a thin slice of the sample and scans a 3D volume [inset (i)]. Emitted fluorescence is collected by the detection objective [inset (ii)]. A high-intensity bleaching laser creates a bleaching volume by performing 3D point-scanning at the center of the 3D illuminated volume [inset (iii)]. $\lambda/2$, half waveplate; PBS, polarizing beamsplitter; S, shutter; 2D GS, 2D galvanometer system; MM, MEMS mirror; IO, illumination objective; DO, detection objective; P, piezo stage; DIC, dichroic mirror; LPF, low-pass filter; T, tube lens; $\lambda_{ill}$, illumination laser; $\lambda_{det}$, detected emission fluorescence; $\lambda_{ble}$, bleaching laser; **b** LiFT-FRAP data collection and analysis workflow. In a LiFT-FRAP experiment, prebleaching images are first recorded, followed by the photobleaching process. Postbleaching images are collected instantly after bleaching. Time series of 3D LiFT-FRAP image data that record the 3D fluorescence recovery process was processed and then converted to the frequency domain through a 3D spatial Fourier transformation. Based on our 3D FRAP theory (Supplementary Note 5), the normalized solute concentration in the frequency domain $\tilde{C}/\tilde{C}_0$ (gray circles) will gradually decrease with the 3D fluorescence recovery ($u, v, w$ are the spatial frequency coordinate. The unit of $u, v, w$ is $\mu m^{-1}$). Fitted with the theoretical equations (red line), the diffusivity value is determined. Then each component of the 3D diffusion tensor can be calculated (Supplementary Note 5).

after CXL treatment, while the structure coherency coefficient was significantly increased in CXL cornea sections than native cornea sections (Fig. 4d). These results demonstrated the changes of tissue structural integrity due to invasive tissue sectioning, which raised concerns about the accuracy of 2D measurements in characterizing the cornea diffusion properties.

Using LiFT-FRAP, we can noninvasively access cornea 3D diffusion properties under physiological conditions. First, we conducted in situ LiFT-FRAP measurements in fresh native porcine eyeballs (Supplementary Fig. 6c and Supplementary Movie 4) to examine whether the extracellular diffusion in native corneas is isotropic or anisotropic. LiFT-FRAP results revealed that the in situ diffusion of 20 kDa FD (FD20) in native corneas was indeed highly 3D anisotropic, with the slowest diffusion along the anteroposterior direction and no significant difference between the mediolateral and superoinferior directions (Fig. 4e). Such 3D diffusion anisotropy is correlated with the anisotropic collagen lamellae structure of cornea tissues, with the lamellae layers stacked in the anteroposterior direction, showing that solutes diffuse more slowly in the direction perpendicular to the lamellae structure (Fig. 4c and Supplementary Fig. 6b). The direction of fibers within adjacent cornea lamellae are largely perpendicular to each other[42,43], so we can assume the overall fiber alignment in the transverse plane to be random within the imaging volume, with comparable diffusion along with the mediolateral and superoinferior directions parallel to the lamellae layer. We also performed in situ LiFT-FRAP measurements in CXL porcine eyeballs to identify the exact impact of CXL treatment on cornea diffusion properties and thus tissue function (Supplementary Fig. 6d). LiFT-FRAP results showed that overall 3D diffusion of FD20 was slower but remained anisotropic in CXL corneas, with the slowest diffusion in the anteroposterior direction (Fig. 4e). This result was expected, for CXL corneas exhibited lamellae structure similar to native corneas, as shown in SHG images (Fig. 4c and Supplementary Fig. 6b). Yet surprisingly, we found that the magnitude of 3D diffusion anisotropy was significantly lower in CXL corneas than native corneas (Fig. 4f) while the structural anisotropy showed no difference between CXL and native corneas (Fig. 4d). The diffusivities after CXL treatment decreased more dramatically in the mediolateral and superoinferior directions (parallel to the lamellae) compared to the anteroposterior direction (perpendicular to the lamellae). This difference might result from the more compact lamellae in CXL corneas, as evidenced by their decreased cornea thickness (Supplementary Fig. 6e, f). This compactness could impose greater impedance to molecular diffusion in directions parallel to the lamellae, reducing the difference among the diffusivities and resulting in decreased diffusion anisotropy in CXL corneas.

In contrast, 2D FRAP measurements in sectioned tissues found no significant difference in overall diffusivity between native and CXL groups (Fig. 4g). The solute diffusivity was also much slower than the value measured by LiFT-FRAP, partially owing to the

use of viscous dextran solution for the prevention of cornea sections from tissue swelling. Additionally, although anisotropic diffusion was found in both native and CXL groups (Figs. 4g), 2D FRAP results showed that the diffusion anisotropy is slightly higher in CXL sections than native sections (Fig. 4h), which is opposite from the significantly decreased diffusion anisotropy seen in LiFT-FRAP results (Fig. 4f). Similar results were also obtained from 2D FRAP measurements with another molecule, sodium fluorescein, showing no significant changes in overall diffusivity but significantly increased diffusion anisotropy in CXL sections (Fig. 4i, j). The diffusion anisotropy changes after CXL treatment measured through 2D FRAP (Fig. 4h, j) followed the same trend of structural anisotropy changes measured from cornea sections (Fig. 4d), which suggests the structural changes caused by the invasive sectioning of the 2D approach could interfere with the detection of the actual diffusion changes in vivo.

To determine how the invasive 2D FRAP measurements impact the cornea in vivo physiology, a computational model based on human cornea geometry was simulated (Fig. 5a and Supplementary Note 1). Results showed that a much lower molecular concentration was predicted and a longer time was needed to reach a certain molecular concentration at the posterior cornea when the 2D FRAP results were applied to simulate the molecular distribution (Fig. 5b–d). Moreover, the use of 2D FRAP results downplayed the impact of CXL on the molecular distribution in cornea tissues. With the LiFT-FRAP results, we can clearly see the difference in the molecular distribution between native and CXL corneas (Fig. 5b–d). However, very minor CXL influence on the concentration profile was found with the 2D FRAP results (Fig. 5b–d), which could lead to misunderstanding of treatment efficacy. These results demonstrate that 2D FRAP measurement on dissected tissue sections is not a reliable approach to recapitulate the 3D biomolecular transport in cornea tissues in vivo. Using LiFT-FRAP, we quantified the baseline in situ diffusion properties in cornea tissues, therefore addressing the long-standing questions regarding molecular transport behavior in cornea tissues. With the ability to characterize the effects of CXL on cornea 3D diffusion properties, LiFT-FRAP provides a key dimension beyond biomechanics for evaluating CXL safety and efficacy, and crucial insight into further improving clinical outcomes. Our results further highlighted the necessity of noninvasive 3D diffusion measurement tools, and the power of LiFT-FRAP to elucidate tissue physiology, pathophysiology, and intervention outcomes.

**LiFT-FRAP quantifies 3D diffusion changes in a tendon disease model.** Applying LiFT-FRAP to quantify the 3D diffusion behavior in biological tissues provides a more thorough understanding of the tissue pathophysiology. Currently, pathophysiology studies are mainly focused on the changes in tissue structures or mechanical properties. The investigation on 3D diffusion lags

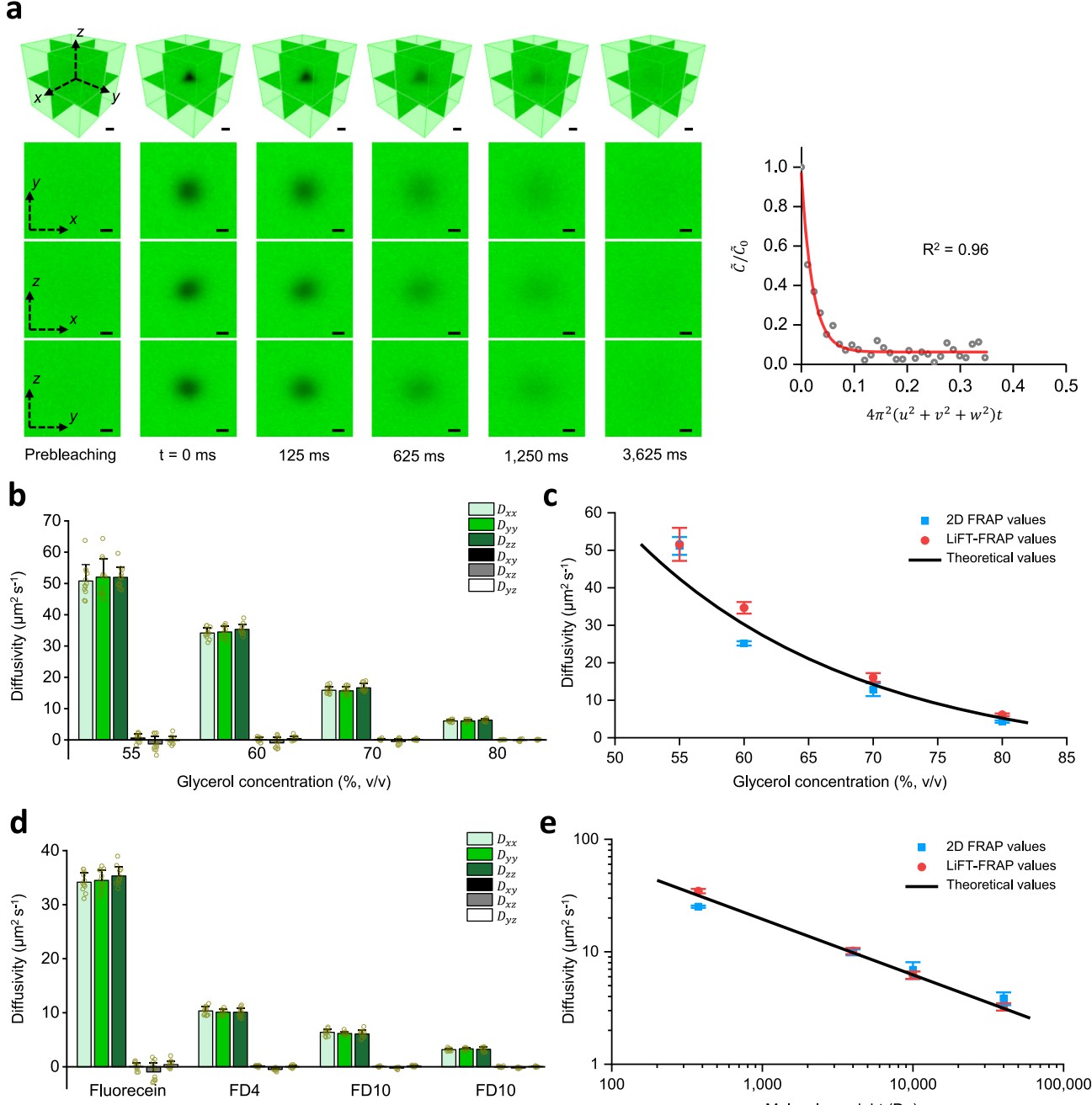

**Fig. 3 Performance of LiFT-FRAP. a** Time series of LiFT-FRAP 3D images of sodium fluorescein in 55% glycerol solution and the corresponding curve fitting results. In a LiFT-FRAP experiment, prebleaching images are first recorded, followed by the photobleaching process. Then postbleaching images are collected. The 3D image acquisition rate is 8 volumes per second. $\tilde{C}/\tilde{C}_0$ is the normalized solute concentration in the frequency domain (gray circles). $u$, $v$, $w$ are the spatial frequency coordinate. The fitted curve is shown in red. Scale bar, 10 μm. **b** 3D diffusion tensor results of sodium fluorescein in different glycerol solutions ($n = 12$ independent measurements). **c** Comparison of 2D FRAP results ($n = 12$ independent measurements), LiFT-FRAP results ($n = 12$ independent measurements), and theoretical results (Supplementary Table 2) for different glycerol solutions. **d** 3D diffusion tensor results of sodium fluorescein and FD molecules in 60% glycerol solutions (n = 12 independent measurements). **e** Comparison of 2D FRAP results ($n = 12$ independent measurements), LiFT-FRAP results ($n = 12$ independent measurements), and theoretical results (Supplementary Table 3) for different sized molecules. 2D FRAP and LiFT-FRAP values in **c** and **e** are the average values of the diagonal components of 2D and 3D diffusion tensor, respectively. All data depict mean ± standard deviation. Source Data is available as a Source Data file for **a**–**e**.

behind mechanical studies even though it is well known that molecular diffusion is important to tissue functions. In this application, we characterized the 3D diffusion of a small probe molecule, sodium fluorescein (376 Da), in both healthy and pathological rat tail tendons (Supplementary Fig. 7a and Supplementary Movie 5). Our results showed that the 3D diffusion of

sodium fluorescein in healthy tendons was strongly anisotropic, with the fastest diffusion along the fiber direction (Fig. 6a, b). These results are consistent with our recent study showing 3D anisotropic diffusion in porcine tendons but with a larger probe, 70 kDa FD (FD70), and much slower extracellular diffusivity[25]. To quantify the 3D diffusion response to pathological tissue

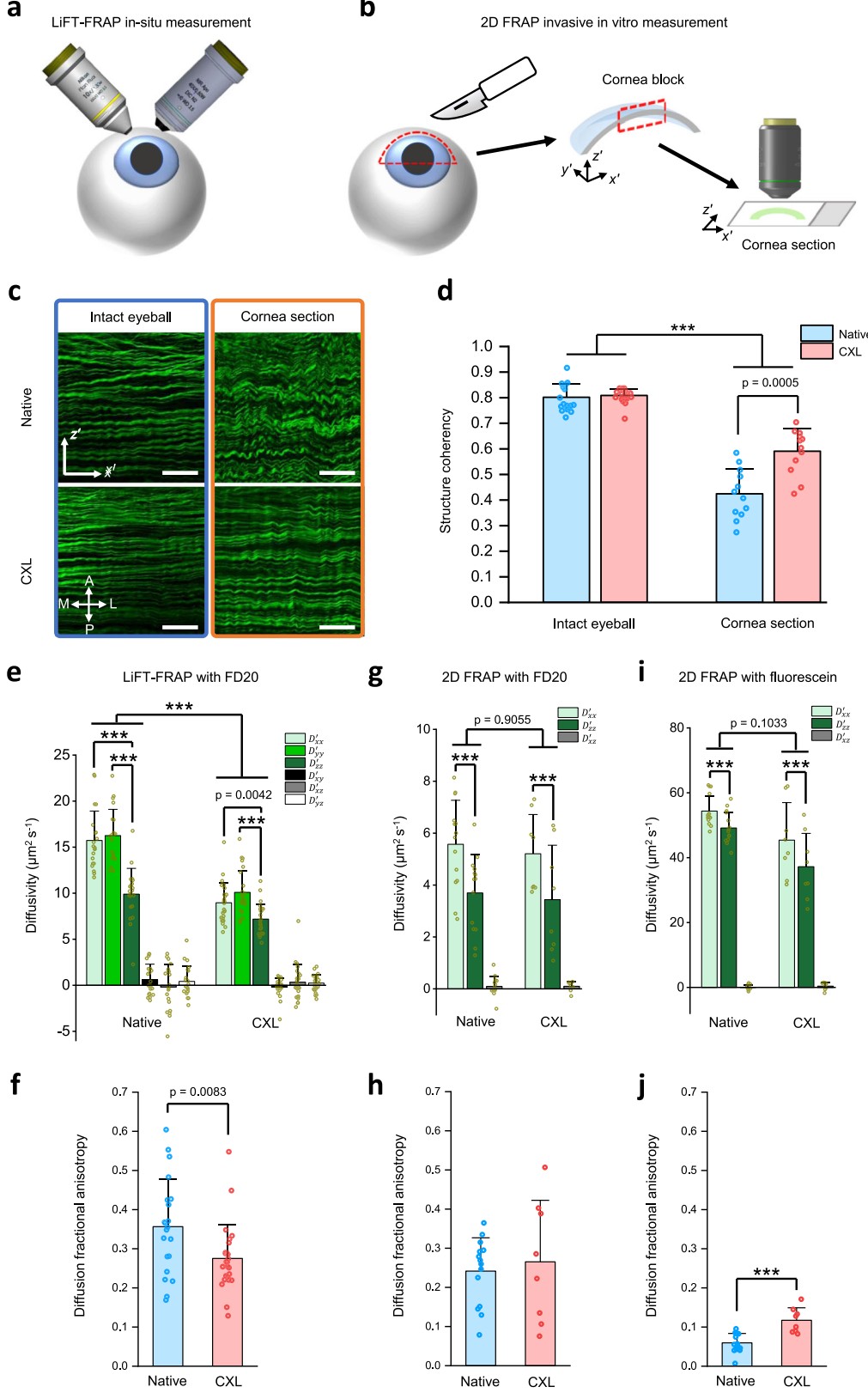

changes, we applied LiFT-FRAP to a thermal treatment model (Supplementary Fig. 7b), as it has been reported that thermally treating collagenous tissues at temperatures over 65 °C will start to denature the collagen fibers[44], causing pathology-mimicking changes in tissue composition and structure. We treated fresh tendons at 70 °C during the sodium fluorescein staining process and found that the tendons were partially denatured after a period of 5 min and fully denatured for 30 min, as demonstrated by SHG imaging and picrosirius red staining with and without polarized light (Fig. 6b, c). Our results showed that the diffusion of sodium fluorescein was highly anisotropic in healthy tendons, but isotropic in both thermally treated tendons (Fig. 6a).

**Fig. 4 Noninvasive 3D LiFT-FRAP measurement versus invasive 2D measurement in cornea tissues. a** Schematic of LiFT-FRAP in situ measurement in the cornea of porcine eyeballs. **b** Schematic of 2D FRAP invasive measurement in the cornea sections. Cornea tissues were first dissected from porcine eyeballs and then cut into thin sections for 2D FRAP measurement. $x'-y'-z'$ is the cornea coordinate system. **c** Representative 2D SHG images of native and CXL corneas in intact eyeballs and cornea sections. Scale bar, 50 μm. A, anterior; P, posterior; M, medial; L, lateral. **d** Structure coherency coefficient of collagen fiber alignment in native and CXL corneas based on their 2D SHG images ($n = 19$ measurements from 9 independent native eyeballs, $n = 21$ measurements from 10 independent CXL eyeballs; $n = 12$ measurements from 6 independent native sections, $n = 12$ measurements from 6 independent CXL sections). ***$p < 0.0001$, one-way ANOVA with Bonferroni post-hoc test. **e** 3D diffusion tensor results of FD20 in native ($n = 21$ measurements from 7 independent samples) and CXL ($n = 25$ measurements from 9 independent samples) corneas. Results are presented in the cornea coordinate system through tensor rotation (Supplementary Note 7). ***$p < 0.0001$, one-way and two-way ANOVA with Bonferroni post-hoc test. **f** 3D diffusion fractional anisotropy in native ($n = 21$ measurements from 7 independent samples) and CXL ($n = 25$ measurements from 9 independent samples) corneas. $p$-value was calculated with a two-sided $t$-test. **g** 2D FRAP results of FD20 in native ($n = 14$ measurements from 5 independent samples) and CXL ($n = 8$ measurements from 4 independent samples) cornea sections. ***$p < 0.0001$, two-way ANOVA with Bonferroni post-hoc test. **h** Diffusion fractional anisotropy calculated from 2D FRAP results of FD20 in cornea sections. **i** 2D FRAP results of sodium fluorescein in native ($n = 15$ measurements from 5 independent samples) and CXL ($n = 8$ measurements from 4 independent samples) cornea sections. ***$p < 0.0001$, two-way ANOVA with Bonferroni post-hoc test. **j** Diffusion fractional anisotropy calculated from 2D FRAP results of sodium fluorescein in native ($n = 15$ measurements from 5 independent samples) and CXL ($n = 8$ measurements from 4 independent samples) cornea sections. ***$p < 0.0001$, two-sided $t$-test. All data depict mean ± standard deviation. Source Data is available as a Source Data file for **d–j**.

Compared to healthy tissues, the 3D diffusion anisotropy of thermally treated tissues was significantly decreased as treatment time increased in a dose-dependent manner (Fig. 6d). Correspondingly, the structure coherency coefficient in treated groups was also significantly decreased in a dose-dependent manner (Fig. 6e). In addition, overall probe diffusivity increased as the duration of thermal treatment increased (Fig. 6a), which correlated with increased tendon water content throughout the thermal treatment process (Fig. 6f). These results demonstrate LiFT-FRAP is a robust technique for 3D diffusion measurement in biological tissues. By quantifying 3D diffusion variations in biological tissues caused by pathological composition and structural changes, LiFT-FRAP is expected to provide crucial insights into disease mechanisms at the molecular level.

**LiFT-FRAP allows tunable 3D diffusion properties in tissue engineering scaffolds.** Another application of LiFT-FRAP is measuring 3D diffusion properties in biomimetic scaffolds. Biomimetic scaffolds have shown great promise in the field of tissue engineering, as their structure can be tailored to recreate microenvironments suitable for tissue formation[45–47]. While significant efforts have been devoted to promoting anisotropic mechanical properties of tissue engineering scaffolds, accurate characterization of scaffold 3D diffusion properties is needed to confirm their functional performance and further enhance their biological relevance[48,49]. Moreover, a standard tool for 3D diffusion measurement is in great need to assure quality control in the industrial scaffold fabrication process. LiFT-FRAP was used to quantify the impact of scaffold composition and structure, as well as diffusing molecule size, on 3D diffusion in scaffolds. To test the effect of scaffold composition on 3D diffusion properties, we first measured the 3D diffusion in gelatin hydrogels (Supplementary Fig. 8a). We found that the 3D diffusion of FD70 was isotropic in all hydrogels, and that diffusivity decreased as hydrogel concentration increased (Supplementary Fig. 8b). These results agreed with the previous findings[1], demonstrating that LiFT-FRAP can characterize the influence of scaffold composition on 3D diffusion.

To delineate the relationship between material structure and molecular size on scaffold 3D diffusion, we used LiFT-FRAP to measure the 3D diffusion of two FD molecules with different molecular weights (10 and 20 kDa) similar to essential growth factors (Supplementary Table 1) in two fiber-based scaffolds–one aligned, one random–fabricated by electrospinning (Supplementary Fig. 9a). In fiber-aligned scaffolds (Fig. 7a and Supplementary Fig. 9b, Supplementary Movie 6), both probes exhibited 3D

anisotropic diffusion, with diffusion occurring fastest along the principal fiber direction (Fig. 7b), while in random fiber scaffolds (Fig. 7c and Supplementary Fig. 9c), diffusion of both probes was isotropic (Fig. 7d). The structure coherency coefficient in fiber-aligned scaffolds was much higher than in random fiber scaffolds (Fig. 7e). Consequently, diffusion anisotropy in fiber-aligned scaffolds was much higher (Fig. 7f), supporting the notion that structure anisotropy leads to diffusion anisotropy[10]. These results revealed 3D anisotropic diffusion properties in tissue engineering scaffolds. In assessing the influence of molecular size on scaffold 3D diffusion, FD10 showed faster diffusivity than FD20 in both fiber-aligned and random fiber scaffolds (Fig. 7b, d), as FD10 has a smaller molecular size compared to FD20. The increased diffusion anisotropy observed with FD20 in fiber-aligned scaffolds (Fig. 7f) further suggested that large molecule diffusion is more sensitive to tissue structure due to its higher directional dependence in physical interactions with oriented fibers. These results demonstrate the capacity of LiFT-FRAP for noninvasively measuring 3D diffusion of biorelevant molecules. As a sensitive tool for quantifying the impact of scaffold composition and structure, as well as molecular size, on 3D molecular diffusion in tissue engineering scaffolds, LiFT-FRAP can be used to guide scaffold development to recapitulate functional tissues, as well as to standardize scaffold fabrication and quality.

## Discussion
Current biotransport research in biological tissues lacks a practical tool that can noninvasively measure 3D extracellular diffusion of biorelevant molecules. Here, we developed LiFT-FRAP to address this issue. Our results demonstrate that LiFT-FRAP is a robust technique for accurately and noninvasively measuring 3D diffusion tensors of biorelevant molecules with various sizes in a diverse range of biological systems, including fibrous tissues and tissue-engineered scaffolds, therefore resolving the technical hurdle that prevents our understanding of tissue pathophysiology, the development of clinical treatment for translational applications, and standardization of biofabrication for the biotech industry.

LiFT-FRAP provides a powerful, noninvasive tool for fast 3D extracellular diffusion measurement. First, LiFT-FRAP can measure 3D diffusion tensors of a wide range of biomolecules. Utilizing our integrated image acquisition system, LiFT-FRAP can achieve 3D diffusion tensor measurements up to 51 μm² s⁻¹, over 100-fold faster than previously reported[25]. This sensitivity now lies in the diffusivity range of most molecules in biological systems (Supplementary Table 1), extensively improving the

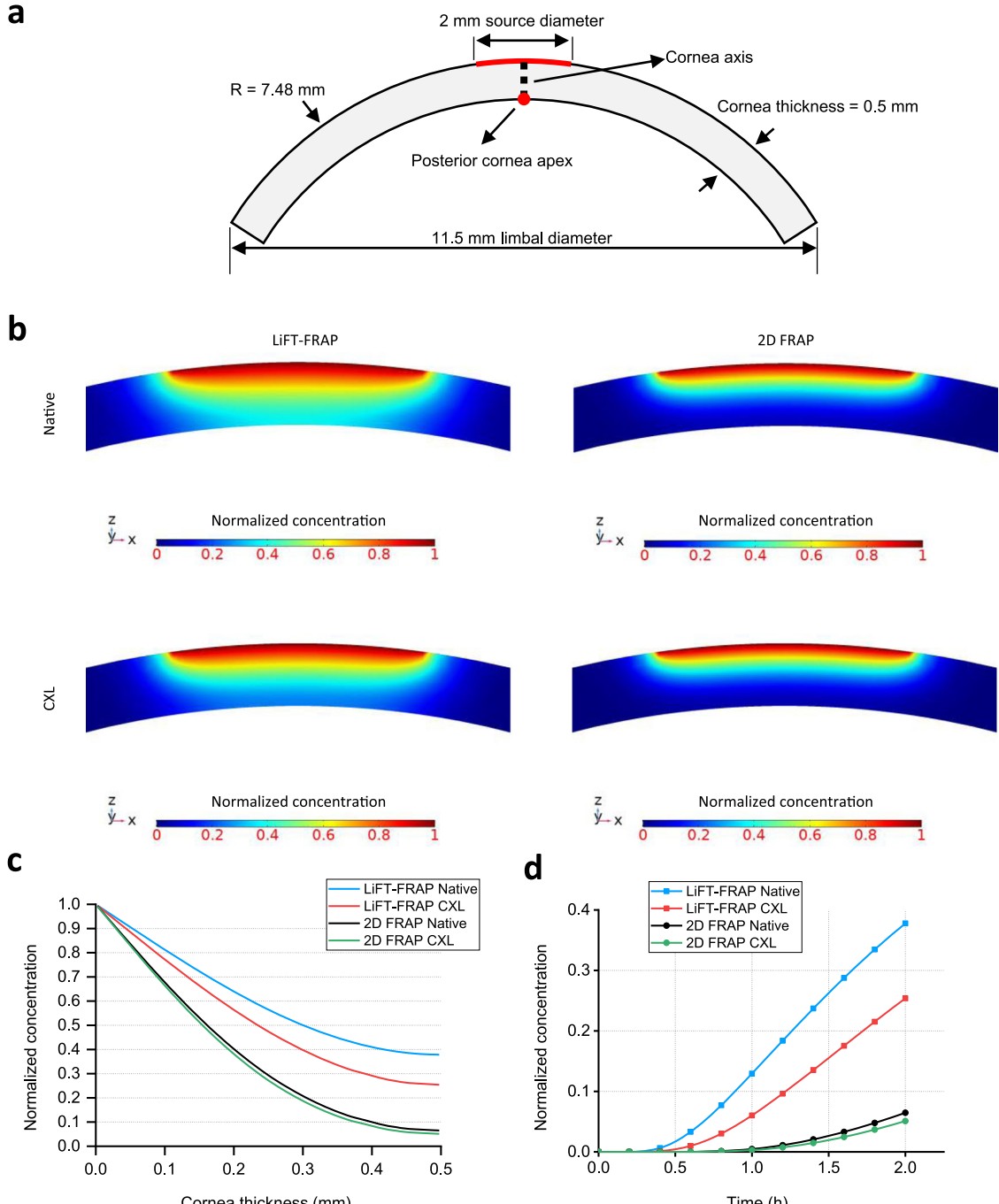

**Fig. 5 Computational modeling of molecular distribution in cornea tissues. a** Cornea geometry for computational simulation experiment. A portion of the anterior cornea with a radius of 1 mm is set as a constant molecular source, which is indicated by the red arch. The black dotted line indicates the cornea axis. The red dot indicates the posterior cornea apex. Zero-flux boundary condition is applied to all of the boundaries in this simulation. **b** Molecular concentration profile within the central region of cornea tissue at a time of 2 h with the diffusivities measured from LiFT-FRAP and 2D FRAP. **c** Molecular concentration profile across cornea axis (black dotted line in **a**) at a time of 2 h with the LiFT-FRAP results or 2D FRAP results as the diffusion input. **d** Molecular concentration at the posterior cornea apex (red dot in **a**) with the LiFT-FRAP results or 2D FRAP results as the diffusion input. Source Data is available as a Source Data file for **c**, **d**.

applicability and impact of LiFT-FRAP in the field of biomedical research. Compared to current technologies used for 3D diffusion tensor measurement, such as MRI-based diffusion tensor imaging[27] and diffusion tensor optical coherence tomography[11], LiFT-FRAP can now measure the 3D diffusion of a wide range of biomolecules beyond water molecules or scattering nanoparticles. This versatility renders LiFT-FRAP a widely applicable tool for 3D diffusion studies in biomedical research. Second, LiFT-FRAP

successfully implements 3D FRAP with two-photon light-sheet microscopy. Although one-photon light-sheet microscopy was very recently introduced into a FRAP system for fast diffusion measurement, it was limited to measuring only 2D isotropic diffusion in thin tissue slices[50,51]. Instead, LiFT-FRAP combines two-photon scanned light-sheet microscopy and a two-photon 3D volume bleaching unit for 3D deep tissue measurement. By separating the optical paths for 3D volume bleaching and

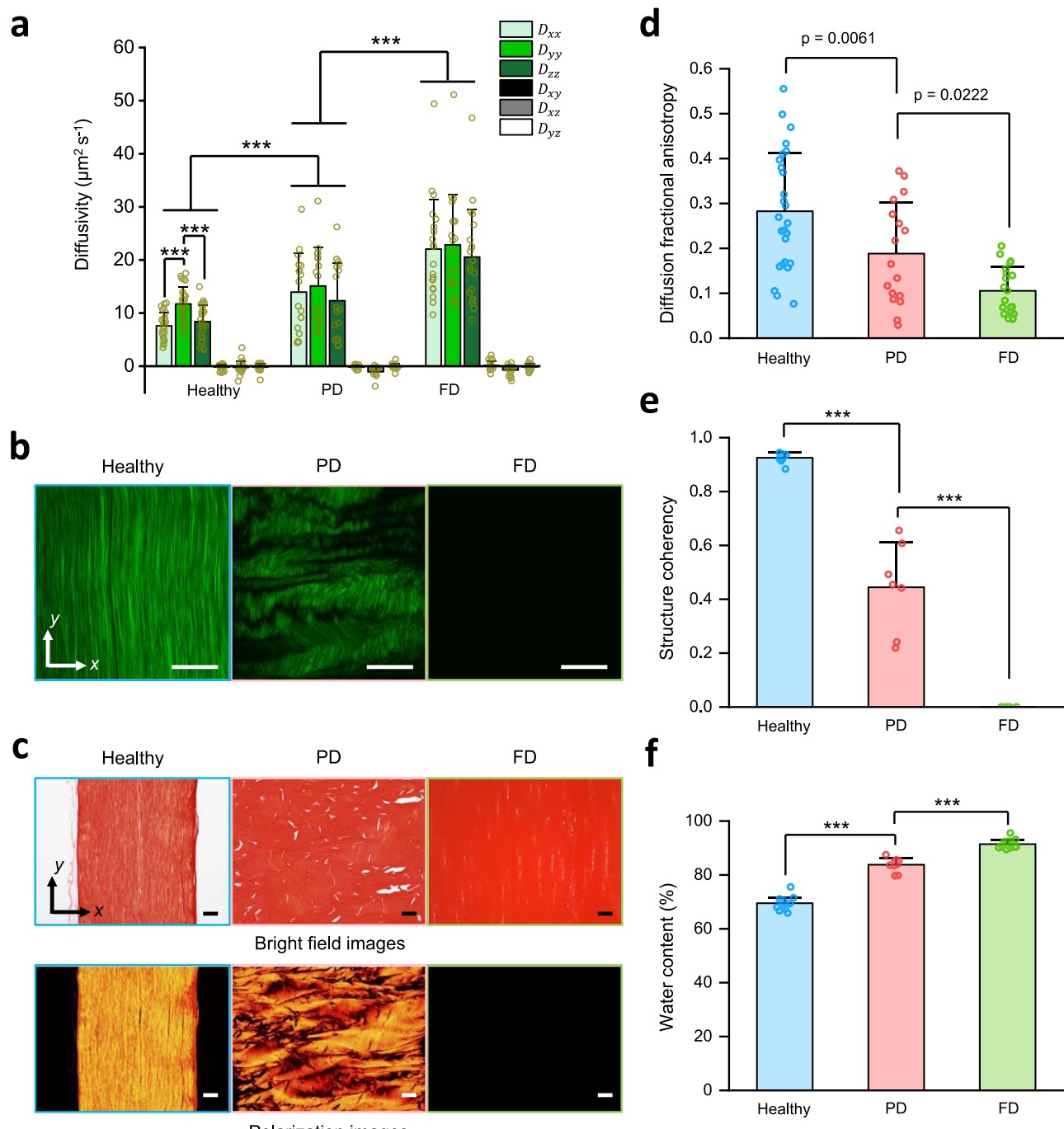

**Fig. 6 Application of LiFT-FRAP in a tendon disease model. a** 3D diffusion tensor results of sodium fluorescein in healthy ($n = 27$ measurements from 10 independent samples), partially denatured (PD) ($n = 17$ measurements from 6 independent samples), and fully denatured (FD) ($n = 20$ measurements from 9 independent samples) tendons. ***$p < 0.0001$, one-way and two-way ANOVA with Bonferroni post-hoc test. **b** Representative SHG image of healthy, PD, and FD tendons at a depth of 150 μm beneath the tissue surface. **c** Representative bright field and polarization images of picrosirius red-stained sections of healthy, PD, and FD tendons. **d** 3D diffusion fractional anisotropy of sodium fluorescein diffusion in healthy ($n = 27$ measurements from 10 independent samples), PD ($n = 17$ measurements from 6 independent samples), and FD ($n = 20$ measurements from 9 independent samples) tendons. $p$-value was calculated with one-way ANOVA with Bonferroni post-hoc test. **e** Structure coherency coefficient of collagen fiber alignment in healthy ($n = 9$ independent samples), PD ($n = 7$ independent samples), and FD ($n = 7$ independent samples) tendons based on their SHG images. ***$p < 0.0001$, one-way ANOVA with Bonferroni post-hoc test. **f** Water content in healthy ($n = 18$ independent samples), PD ($n = 9$ independent samples), and FD ($n = 17$ independent samples) tendons. ***$p < 0.0001$, one-way ANOVA with Bonferroni post-hoc test. All data depict mean ± standard deviation. All scale bars represent 50 μm. Source Data is available as a Source Data file for **a**, **d**–**f**.

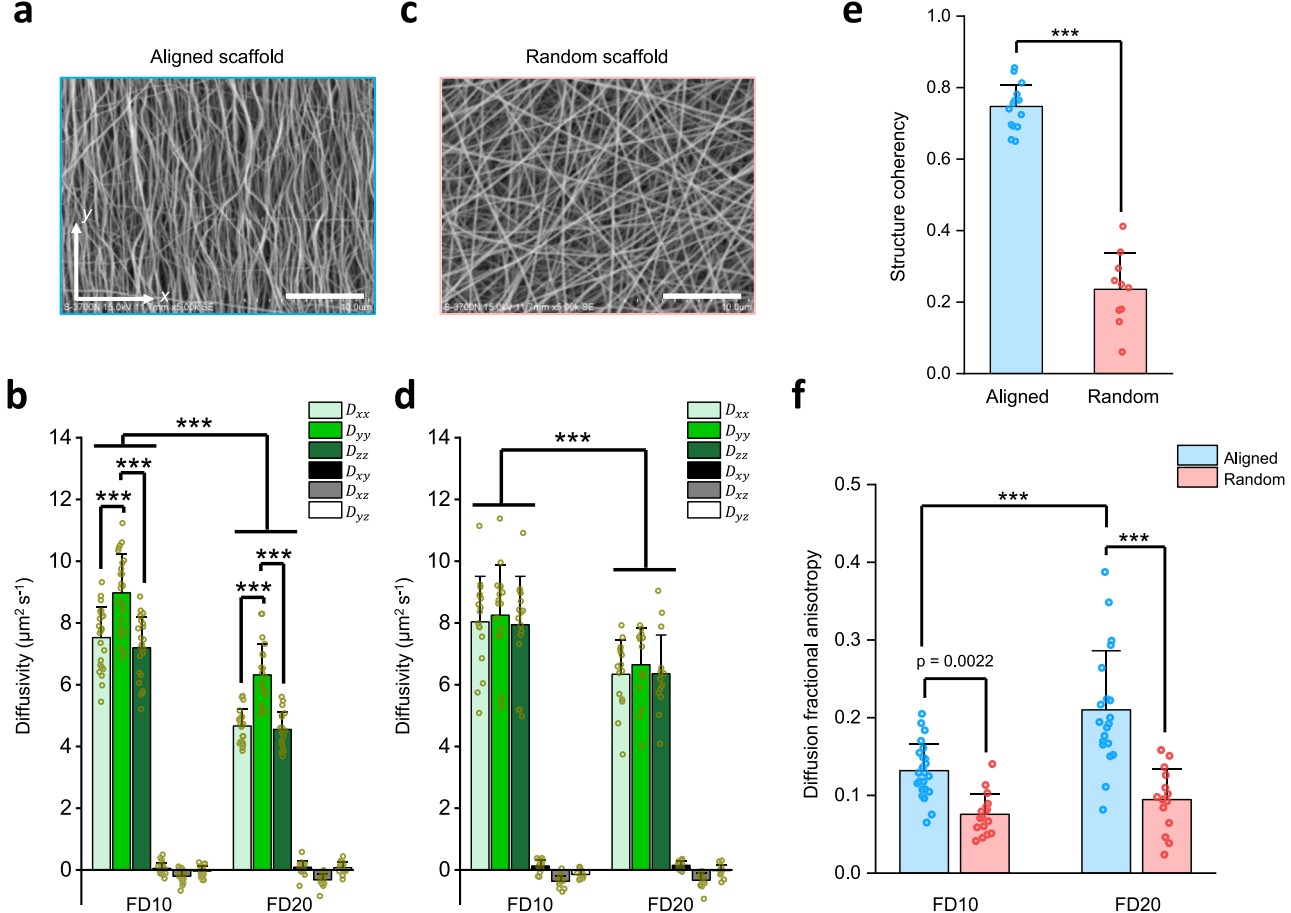

**Fig. 7 Application of LiFT-FRAP in tissue engineering scaffolds. a** Scanning electron microscopy (SEM) image of the fiber-aligned scaffold. **b** 3D diffusion tensor results of FD10 and FD20 in fiber-aligned scaffolds ($n = 24$ measurements from 8 independent scaffolds for FD10 and $n = 20$ measurements from 7 independent scaffolds for FD20). ***$p < 0.0001$, one-way and two-way ANOVA with Bonferroni post-hoc test. **c** SEM image of random fiber scaffold. **d** 3D diffusion tensor results of FD10 and FD20 in random fiber scaffolds ($n = 17$ measurements from 6 independent scaffolds for FD10 and $n = 15$ measurements from 5 independent scaffolds for FD20). ***$p < 0.0001$, two-way ANOVA with Bonferroni post-hoc test. **e** Structure coherency coefficient of fiber alignment in aligned ($n = 16$ independent samples) and random ($n = 10$ independent samples) scaffolds based on their SEM images. ***$p < 0.0001$, two-sided $t$-test. **f** 3D diffusion fractional anisotropy calculated from 3D diffusion tensor results of FD molecules in fiber-aligned scaffolds ($n = 24$ measurements from 8 independent scaffolds for FD10 and $n = 20$ measurements from 7 independent scaffolds for FD20) and random fiber scaffolds ($n = 17$ measurements from 6 independent scaffolds for FD10 and $n = 15$ measurements from 5 independent scaffolds for FD20). ***$p < 0.0001$, one-way ANOVA with Bonferroni post-hoc test. All data depict mean ± standard deviation. Both scale bars represent 5 μm. Source Data is available as a Source Data file for **b**, **d**–**f**.

light-sheet illumination, our LiFT-FRAP system facilitates fast switching between bleaching and postbleaching imaging procedures. Moreover, synchronizing the movements of the light sheet with the detection objective instead of translating the sample enables faster and more stable 3D image acquisition.

LiFT-FRAP achieves the long-standing goal of the FRAP technique by expanding the scope of diffusion studies from 2D to 3D space. Since FRAP was originally developed in the 1970s, it has been mainly used for isotropic diffusion measurement[15,28,30–34]. Recently, a number of FRAP methods have been developed to attempt to investigate 3D anisotropic diffusion in tissues[10,20,35–38]. However, these methods are 2D-based, neglecting diffusion along optical axis[10,20,21,23,35–37]. Intuitively, we should be able to reconstruct the 3D extracellular diffusion through multiple 2D measurements in the 3D orthogonal planes of tissue samples. However, our results demonstrate that such a strategy suffers from major limitations. First, invasive tissue dissection disrupts tissue structure, as shown with the cornea (Fig. 4c, d). Second, excised tissue sections are prone to swelling due to the loss of tissue integrity and

intrinsic mechanical environment. Although the introduction of a high-osmotic solution, such as the dextran solution used for the 2D measurements in cornea tissues, can alleviate the swelling issue, it may alter tissue diffusion properties due to possible tissue shrinkage and infiltration of the viscous solution. Therefore, the diffusion results measured by this strategy cannot represent the actual diffusion properties under physiological conditions in vivo, limiting the ability to identify the diffusion response to diseases or treatments. LiFT-FRAP, instead, enables the determination of a complete 3D diffusion tensor from a single FRAP experiment, thereby allowing noninvasive quantification of diffusion in any direction within 3D tissue samples and thus resolving all the issues 2D methods encountered. In addition, LiFT-FRAP accommodates flexible initial conditions of the bleaching region, allowing the bleaching spot to assume any shape, and diffusion to take place during the bleaching process. This enhances the accuracy and convenience of FRAP measurement[25,38].

LiFT-FRAP was successfully applied to study the 3D diffusion properties in cornea tissues. The result confirmed the diffusion in

cornea tissues is highly anisotropic under physiological conditions, which is correlated with the anisotropic cornea fiber structures. This finding agrees with previous in situ MRI-based diffusion tensor imaging results[26] but disagrees with the fluorescence correlation spectroscopy reports[24]. The discrepancies may be attributed to the excision procedures involved in the latter study. As demonstrated in our 2D FRAP measurements in cornea sections, the invasive procedure trends to decrease tissue diffusion anisotropy. LiFT-FRAP was also used to quantify the CXL effects on cornea 3D diffusion properties. Current CXL studies mainly focus on the influence of CXL on cornea mechanical properties[52,53], but its effects on 3D diffusion properties are unknown. Our results showed that CXL decreases both molecular diffusivities and diffusion anisotropy. Considering the importance of diffusion in defining the biochemical environment necessary for maintaining normal function in biological tissues, quantification of 3D diffusion changes caused by CXL could be critical for understanding the CXL impact on long-term tissue health. Additionally, we found no changes in cornea structure anisotropy but a significant decrease in cornea thickness after CXL treatment, which indicates morphological structure analysis based on fiber alignment alone is not sensitive enough to reveal the CXL effect on cornea tissues. Functional analysis, like the 3D diffusion measurement in cornea tissues through LiFT-FRAP, is a sensitive means to detect the overall tissue changes caused by the CXL procedure. LiFT-FRAP enables accurate measurement of both the baseline diffusion properties in native tissues and the diffusion tensor in treated tissues, therefore providing a valuable method to improve treatment safety and efficacy.

Applying LiFT-FRAP to biological tissues, we found that 3D anisotropic diffusion is prevalent in physiological tissues but altered in diseased tissues. Changes in tissue structure or composition were shown to impact 3D diffusion behavior and thus the underlying biochemical environment. These results indicate that 3D anisotropic diffusion might play an important role in supporting tissue functions, as suggested in previous studies wherein 3D anisotropic diffusion was concluded to facilitate the transport of biomolecules, such as glutamate[54]. Many studies have since correlated tissue dysfunction with loss of tissue structure or mechanical properties[55–57], yet very few have been able to investigate the effects of pathological structure and function changes on diffusion and the larger tissue biochemical environment. LiFT-FRAP can now accomplish this feat by measuring the 3D diffusion of various biomolecules at different stages along with the disease progression, allowing us to quantitatively assess changes in tissue biochemical environment and elucidate disease mechanisms.

By enabling accurate quantification of 3D diffusion behavior in scaffolds, LiFT-FRAP serves to augment current tissue engineering approaches. Like endogenous tissues, biomimetic scaffolds must possess several types of highly direction-dependent organization, including structural, mechanical, and diffusion anisotropy[49]. Although numerous studies have been devoted to mimic the structural and mechanical anisotropy of native tissues, diffusion anisotropy has not been well studied due to the lack of tools capable of examining 3D diffusion in tissues and scaffolds[45–49]. Now that LiFT-FRAP can accomplish 3D diffusion tensor measurements in both systems, new scaffold structures can be designed and evaluated based on their 3D diffusion properties to more closely recapitulate endogenous tissue diffusion anisotropy.

While two-photon excitation allows diffusion measurement deep into natively transparent tissues like cornea tissues, for denser tissues, optical clearing agents, such as the glycerol solutions used in our rat tail tendon experiments, are needed to reduce tissue light scattering. However, glycerol introduction

could potentially result in tissue dehydration[58] and alter tissue diffusion properties. Therefore, future studies will be designed to investigate 3D diffusion in highly scattering tissues without optical clearing agents. The use of near-infrared dye, with both its excitation and emission spectra inside the near-infrared window, might be one such option[59–61]. Alternative methods to improve imaging depth, such as microfiberoptic epifluorescence photobleaching[1] or photoacoustic recovery after photothermal bleaching[62], may also be considered. In addition, although here we only demonstrated the combination of light-sheet microscopy with 3D Fourier transform FRAP, other fast 3D imaging techniques, such as light-field microscopy[63], might also be considered to further evolve FRAP-derived techniques for 3D diffusion measurement. The ability of LiFT-FRAP to discriminate between different models of diffusion, such as anomalous diffusion, was not explored in this study. However, given the successful demonstration of the spatial Fourier analysis method for simultaneous measurement of both normal diffusion and binding rate in 2D[64,65], our method could be extended to quantify more complex 3D diffusion behaviors in the future.

By demonstrating the prevalence of 3D anisotropic diffusion in biological systems and the direction-dependency of diffusion responses to tissue structure changes, LiFT-FRAP provides a powerful platform technique for noninvasive 3D diffusion tensor measurement with a diverse range of biomedical applications, capable of advancing clinical outcomes, studying disease mechanisms, and improving the characterization and functional optimization of tissue engineering scaffolds. With the commercialization of 3D light-sheet microscopy, we expect that LiFT-FRAP will become an accessible and practical tool with broad applications in biomedical research.

## Methods

**LiFT-FRAP setup.** The LiFT-FRAP system, illustrated in Fig. 2a and Supplementary Fig. 1, consists of three functional units: (i) a light-sheet unit that generates a two-photon excitation light sheet to illuminate a 3D volume, (ii) a detection unit that records emitted fluorescence with a high-speed camera, and (iii) a two-photon volume bleaching generator that creates a 3D bleaching volume with a raster scan of the ultrafast laser beam. The system used an ultrafast laser (Tsunami HP, Spectra Physics, Mountain View, CA) tuned to 800 nm as the light source. The output of the laser was split into two beams by a broadband tunable laser beam splitter composed of a half waveplate (10RP52-2B, Newport, Irvine, CA) and a polarizing beam splitter (05FC16PB.5, Newport, Irvine, CA). The two beams were then fed into the light-sheet unit and the volume bleaching generator respectively. Each laser beam was turned on and off by an electric laser shutter (SR474, Stanford Research System, Sunnyvale, CA). In the light-sheet unit, the illumination laser was extended along the $x$ axis (Supplementary Fig. 3a) by partially filling the back aperture of the illumination objective (N10XLWD-NIR, ×10 NA 0.3, Nikon, Japan). A 2D gimbal MEMS scanning mirror (gold coated, integrated, diameter 800 μm, Mirrorcle, Richmond, CA) scanned the extended laser along the $y$ axis to generate a light sheet (Supplementary Fig. 3b) and along the $z$ axis to achieve 3D illumination. In the volume bleaching generator, the bleaching laser was reflected by a dichroic mirror (FF662-FDi01, Semrock, Rochester, NY) and focused onto the center of the illuminated 3D volume by the detection objective (N40XLWD-NIR, ×40, NA 0.8, Nikon, Japan). This objective was also used in the detection unit for recording fluorescence images. Mounted on a piezo stage (PD72Z4CAQ, Z-mover, 400 μm, PI, German), the detection objective was driven along the $z$ axis to allow continuous volumetric bleaching or imaging. To create a 3D bleaching volume, a 2D galvanometer system (GVS002 dual-axis, Thorlabs, Newton, NJ) scanned the bleaching laser in the $xy$ plane with an area of 10 μm × 10 μm, and the piezo stage drove the movement of detection objective in the $z$ axis by 10 μm. In the detection unit, the emitted fluorescence from the light-sheet illumination plane was captured by the detection objective, then passed through the dichroic mirror and a low-pass filter (FF01-665/SP-25, Semrock, Rochester, NY), and was finally focused onto a high-speed sCMOS camera (Zylar 4.2 sCMOS, Andor, UK) via the tube lens (EF 28–135 mm, Canon, Japan). Images had a size of 192 pixels × 192 pixels (76 μm × 76 μm) and were recorded at 624 frames per second. The 3D imaging was implemented by moving the detection objective along the $z$ axis in a range of 80 μm while keeping the sample static. Each 3D stack consists of 78 frames, resulting in a 3D imaging rate of 8 volumes per second. During 3D imaging, the light-sheet illumination plane was always kept in the focus of the detection objective. Operations of the MEMS mirror, piezo stage, shutters, 2D galvanometer, and

camera acquisition were conducted by the control software written in LabVIEW 2014 32-bit (National Instruments, Austin, TX) (Supplementary Note 2).

**LiFT-FRAP data acquisition and analysis**. In each LiFT-FRAP experiment, 3D prebleaching images were first recorded at a rate of 8 volumes per second for 10 s. The dimension of each LiFT-FRAP 3D volume is 76 μm × 76 μm × 76 μm, consisting of 64 images with a size of 192 pixels × 192 pixels. Then, the central region of the 3D observed volume with a size of 10 μm × 10 μm × 10 μm was photobleached by the high-power bleaching laser for 1 s. Finally, up to 80 volumes of 3D postbleaching images were captured at the same rate as the prebleaching images, 8 volumes per second. LiFT-FRAP experimental settings for each sample type in this study are listed in Supplementary Table 4. All of the laser power values were measured right after the objective with a power meter. No photodamage of tissue samples was observed in any of the experiments. The influence of local heating caused by the bleaching laser illumination on the molecular diffusion measurement is negligible within the bleaching laser power range used in this study (Supplementary Fig. 10). To obtain images with homogenous background, each postbleaching image was normalized by its corresponding prebleaching image (Supplementary Fig. 4 and Supplementary Note 4). The normalized LiFT-FRAP images were then used to

calculate the 3D diffusion tensor $\mathbf{D} = \begin{bmatrix} D_{xx} & D_{xy} & D_{xz} \\ D_{xy} & D_{yy} & D_{yz} \\ D_{xz} & D_{yz} & D_{zz} \end{bmatrix}$ based on our 3D spatial

Fourier transform FRAP theory[25] (Fig. 2b and Supplementary Note 5). A custom MATLAB code (MATLAB 2017a, The MathWorks Inc., Natick, MA) was written to perform the data analysis. All LiFT-FRAP experiments were performed at room temperature (20 °C). 3D images were displayed with Imaris (Bitplane, UK). Movies were generated with Adobe Photoshop (Adobe, San Jose, CA). Data graphs were plotted with Origin (OriginLab, Northampton, MA).

**3D diffusion anisotropy quantification**. To quantify the 3D diffusion anisotropy of fluorescent molecules in scaffolds and biological tissues, diffusion fractional anisotropy (FA) was defined as[66],

$$\text{FA} = \frac{\sqrt{\left[(\lambda_1 - \lambda_2)^2 + (\lambda_1 - \lambda_3)^2 + (\lambda_2 - \lambda_3)^2\right]}}{\sqrt{2[\lambda_1^2 + \lambda_2^2 + \lambda_3^2]}} \quad (1)$$

where $\lambda_1$, $\lambda_2$, and $\lambda_3$ are the principal diffusivities of the 3D diffusion tensor. To calculate the diffusion anisotropy from 2D FRAP results, $\lambda_3$ was assumed to be identical to $\lambda_2$, the larger principal diffusivity of the 2D diffusion tensor. Diffusion fractional anisotropy is a scalar value in a range of zero to one, where a value of zero indicates pure isotropic diffusion, while a value of one indicates pure uni-directional anisotropic diffusion.

**LiFT-FRAP sample preparation**. *Solution samples*: Four different 0.1 mM sodium fluorescein (376 Da, Fluka-Sigma-Aldrich, St. Louis, MO) glycerol/phosphate-buffered saline (1× PBS) (Sigma, St. Louis, MO) solutions were prepared at four different glycerol concentrations (v/v: 55%, 60%, 70%, 80%). Three unique 10 mg mL⁻¹ fluorescein isothiocyanate-dextran (FD) 60% glycerol/PBS solutions were made with three different fluorescent molecules (4, 10, 40 kDa FD, Sigma, St. Louis, MO) (FD4, FD10, and FD40). A total of 25 μL of these fluorescent glycerol solutions were pipetted into square glass tubes (inner dimension: 0.8 mm × 0.8 mm × 50 mm, wall thickness: 0.16 mm) (#8280, VitroCom, Mountain Lakes, NJ), and both ends of the glass tube were immediately sealed with soft silicone putty to prevent leakage and convection.

*Porcine cornea samples*: Fresh porcine (Yorkshire, 6 months) eyeballs were obtained from a local abattoir and used within 12 h post-mortem to determine the 3D diffusion tensor of FD20 in corneas. The use of animal tissues is approved by Clemson University and the Medical University of South Carolina. Eyeballs were stored in moist chambers with PBS-soaked gauze at 4 °C until use. The cornea epithelial layer of each eyeball was completely removed with a blunt Hockey knife. To determine the impact of collagen crosslinking (CXL), an FDA-approved medical procedure for keratoconus patients[39], on cornea diffusion properties, two types of porcine corneas were prepared: native corneas and CXL corneas. Native corneas were stained with 20 kDa FD (FD20) by sequentially applying 20 mg mL⁻¹ FD20 in 1× PBS solution for 10 min for fast dye infiltration and 20 mg mL⁻¹ FD20 in 20% PBS-based Dextran-T-500 (Sigma, St. Louis, MO) for 15 min to bring cornea thickness back to its initial thickness before staining (Supplementary Fig. 6e). CXL corneas were first treated by a standard epithelium-off CXL protocol[39]. Briefly, drops of 0.1% riboflavin-5-phosphate (Sigma Aldrich, St. Louis, MO) in 20% PBS-based Dextran-T-500 (Sigma, St. Louis, MO) were topically applied to the cornea every 5 min for 30 min. Then, the central part of the cornea was subjected to 3.0 mW ultraviolet A (UVA, 370 nm) light for 30 min. During UVA irradiation, riboflavin drops were added every 5 min. After UVA irradiation, CXL corneas were stained with 20 mg mL⁻¹ FD20, 20% PBS-based Dextran-T-500 solution for another 30 min. Eyeballs were then transferred to a custom specimen holder (Supplementary Fig. 6a), and capped with a thin fluorinated ethylene propylene (FEP) film (50A Dupont Teflon, American Durafilm, Holliston, MA) with a refractive index similar to water[67], to cover the central region of the porcine corneas. Cornea thickness was measured with a pachymeter (Ipac Pachymeter,

Reichert, Depew, NY) along the way of sample preparation. Cornea thickness of the intact eyeballs was referred to as the epithelial-on cornea thickness (Epi-on). Cornea thickness without the epithelial layer was referred to as the epithelial-off cornea thickness (Epi-off). Cornea thicknesses measured immediately after-dye staining were referred to as the after-dye cornea thickness (Aft-dye).

*Rat tail tendon samples*: Fresh rat tails were harvested from Sprague Dawley rats (3 months old, 250 g) at the conclusion of other IACUC approved research projects at the Medical University of South Carolina. The use of animal tissues is approved by Clemson University and the Medical University of South Carolina. The tails were used within 12 h of killing to determine the 3D diffusion tensor of sodium fluorescein in tail tendons. Rat tails were stored in a moisture chamber with PBS-soaked gauze at 4 °C until use. To prepare rat tail tendon specimens, tendon fascicles were excised from tendon bundles and placed inside glass tubes. To investigate the impact of tissue structure and composition changes on 3D diffusion properties, rat tail tendons were thermally treated for different amounts of time. Three types of rat tail tendons were prepared: healthy tendons, partially denatured tendons, and fully denatured tendons. Healthy tendons were stained in 0.1 mM sodium fluorescein glycerol/PBS solution (v/v: 60%) at room temperature for 2.5 h. Partially and fully denatured tendons were immersed in the staining solution at 70 °C for 5 min and 30 min, respectively, before resting in staining solution at room temperature for 2 h.

*Hydrogel sample preparation*: Homogenous gelatin gel was prepared at different concentrations (w/w: 2%, 5%, and 8%), mixed with 70 kDa FD (FD70) (Sigma, St. Louis, MO) to reach a final dye concentration of 10 mg mL⁻¹. FD70 powder and gelatin powder (Gelatin from porcine skin, Sigma, St. Louis, MO) were fully mixed with deionized water and incubated in a 70 °C water bath for 15 min to completely melt the gelatin. Subsequently, 25 μL fluorescent gelatin solution was immediately pipetted into the square glass tubes before resting at room temperature until fully gelled.

*Fiber-based scaffold samples*: Fiber-based gelatin scaffolds were fabricated through electrospinning (Supplementary Fig. 9a) with two different fiber alignments: aligned and random. 200 mg mL⁻¹ gelatin solution was prepared by dissolving gelatin powder (Gelatin from porcine skin, Sigma, St. Louis, MO) in 75% (v/v) acetic acid (Sigma-Aldrich, St. Louis, MO) solution[68]. Gelatin solution was then transferred to a 10 mL syringe with a 23 G needle and pumped out of the 10 mL syringe by 3 mmHg of pressure with a custom hydraulic system. Electrospinning was performed at 14 kV and 60% humidity, with the needle tip 10 cm from the grounded mandrel (diameter = 51 mm, width = 3 mm). For the fabrication of fiber-aligned scaffolds, the grounded mandrel was rotated at 1200 rpm. For the fabrication of random fiber scaffolds, the grounded mandrel was held static. Both scaffolds were collected after 4 h of electrospinning and dried in a petri dish at room temperature for 2 days. To prevent gelatin fibers from dissolving in the water-based staining solution, 3D scaffolds were crosslinked with 200 mM 1-ethyl-3-(3-dimethylaminopropyl)carbodiimide hydrochloride (EDC) (#22980, ThermoFisher Scientific, Waltham, MA) and N-Hydroxysuccinimide (NHS) (#130672, Sigma-Aldrich, St. Louis, MO), 100% ethanol solution overnight[69]. After crosslinking, fiber-based scaffolds with a fiber diameter of ~200 nm were placed in glass tubes and stained in 10 mg mL⁻¹ FD10 or FD20, 40% glycerol/water solution for one day at room temperature.

**2D FRAP measurements**. *Solution samples*: A 0.1 mM sodium fluorescein glycerol solution (glycerol concentrations: 55%, 60%, 70%, and 80%) and 10 mg mL⁻¹ FD solutions (FD4, FD10, and FD40) were prepared following the same protocol used for LiFT-FRAP solution experiments. Fluorescent solutions were placed into a thin well chamber made using a double-side adhesive spacer (9 mm in diameter and 0.12 mm in thickness, Secure-Seal spacers, Life Technologies, Grand Island, NY) on a slide. A coverslip was then used to seal the chamber.

*Porcine cornea samples*: Corneas were first dissected out from both native and CXL porcine eyeballs, sap frozen, and then cut into 120 μm thin cornea sections with a cryostat (CM1510S, Leica Microsystem, Inc., Exton, PA). Cornea sections were carefully placed into a thin well chamber made with a double-side adhesive spacer on a slide. Then fluorescent solutions were immediately added to stain the tissue sections. To determine the diffusion of FD20 in cornea sections, 10 mg mL⁻¹ FD20 in 17.5% PBS-based Dextran-T-500 solution were used. Dextran was used to prevent cornea sections from swelling. For diffusion measurement with fluorescein in cornea sections, 0.2 mM sodium fluorescein in 17.5% PBS-based Dextran-T-500 solution was used. After one-hour incubation, excessive fluorescent solutions were removed, and the well with cornea sections was sealed with a coverslip.

*2D FRAP procedure*: The 2D FRAP experiments were performed on a Leica TCS-SP5 confocal laser scanning microscope (Leica Microsystem, Inc., Exton, PA) with an Ar-488 nm laser and an HC PlanAPO 20x/0.7NA air objective (Leica Microsystem, Inc., Exton, PA)[20]. Briefly, the imaging position was set to 7 μm beneath the coverslip, and five prebleaching images plus 200 postbleaching images (128 pixels × 128 pixels) were recorded at low laser intensity (3–7%). To avoid photobleaching artifacts during the long postbleaching imaging process, imaging settings for slow diffusing molecules were different from fast-diffusing molecules, with frame size varied from 775 μm × 775 μm to 258 μm × 258 μm and frame rate varied from 0.355 to 1.065 s per frame, depending on probe diffusivity. For bleaching, laser intensity was increased to 100% to bleach a square region with a side length of 1/8 of the prebleaching image, located at the image center.

Prebleaching images were averaged and subtracted from postbleaching images, which were then analyzed with a custom MATLAB code to calculate 2D FRAP diffusivities[20].

**SEM imaging**. The fiber structure of the fiber-based scaffolds (aligned and random) was determined by scanning electron microscopy (SEM). Scaffolds collected from the mandrel were coated with a 20 nm gold layer before SEM imaging to improve image contrast. High-resolution SEM images were taken with an S-3700N SEM (Hitachi, Japan) at ×5000 magnification.

**SHG imaging**. Second-harmonic generation (SHG) imaging was performed to determine collagen fiber structure in porcine corneas (native and CXL) and rat tail tendons (healthy and thermally treated). For intact porcine corneas, whole eyeballs with cornea epithelial layer removed were first fixed in 10% formalin solution (Fisher Scientific, Hampton, NH) to preserve the collagen fiber structure[70]. After fixation, cornea tissues were dissected and cut in half along the mediolateral direction with a scalpel blade. The transverse plane of each cornea was placed face down against the glass bottom of a petri dish (Cellvis, Mountain View, CA) for SHG imaging. For cornea sections, cornea tissue was first dissected out from both native and CXL porcine eyeballs, sap frozen, and then cut into 120 μm thin cornea sections with a cryostat. Cornea sections were carefully placed in a glass-bottom petri dish for SHG imaging. For rat tail tendons, fresh glycerol-cleared tendon samples were transferred to a glass-bottom petri dish for SHG imaging. SHG images were taken with a ×60 oil immersion lens (UApo N, NA 1.30, Olympus, Center Valley, PA) on an inverted Olympus FV1200MPE multiphoton laser scanning microscope (Olympus, Center Valley, PA). SHG microscopy excitation wavelength was set to 860 nm, and SHG signal was collected in the 420–460 nm range. For SHG imaging of porcine corneas, 2D images were taken at the surface of the transverse plane while 3D stack SHG images were taken from the surface of the transverse plane down to a depth of 80 μm. The distance between sequential images is 1 μm. For SHG imaging of rat tail tendons, the image plane was set to 150 μm beneath the surface.

**Picrosirius red staining and imaging**. Picrosirius red staining was used to determine collagen fiber structure in porcine corneas (native and CXL) and rat tail tendons (healthy and thermally treated). Porcine corneas and rat tail tendons were fixed in 10% formalin solution, dehydrated, and paraffin-embedded. 5-μm-thick tissue sections were cut, de-waxed in xylene, and dehydrated in graded ethanol solutions. Slides were stained in Sirius red 0.1% in saturated picric acid solution (Electron microscopy sciences, Hatfield, PA) and then imaged with a 20x objective (UPlanSApo, NA 0.75, Zeiss, Germany) on an Olympus BX53 microscope (Olympus, Center Valley, PA). Picrosirius red staining images were captured under polarized light to detect collagen birefringence signal against the dark field of view[71]. Bright-field images were also taken under bright light at the same sample location immediately following polarization imaging.

**Fiber alignment quantification**. Fiber alignments of porcine corneas (native and CXL), rat tail tendons (healthy and thermally treated), and fiber-based scaffolds (aligned and random) were quantified based on 2D SHG images (for porcine corneas and rat tail tendons) and SEM images (for fiber-based scaffolds). An ImageJ (1.52a, National Institutes of Health, Bethesda, MD) plug-in, OrientationJ, was used to determine the coherency coefficient of the fiber structure in all the samples[23]. The structure coherency coefficient is in the range of zero to one, where a coherency coefficient of zero indicates fibers are perfectly randomly distributed without a predominant orientation, while a coherency coefficient of one indicates all fibers are aligned in the same direction.

**Water content measurement**. The water content in rat tail tendons (healthy and thermally treated) was measured and compared to determine tissue composition changes. The wet weight, $w_{wet}$, of each rat tail tendon was first measured by a Sartorius balance (LA 120S, Germany). Subsequently, tissue samples were lyophilized and specimen dry weight, $w_{dry}$, was measured. Water content was calculated by $\frac{w_{wet} - w_{dry}}{w_{wet}} \times 100\%$.

**Statistical analysis**. A one-way ANOVA with Bonferroni post-hoc test was performed to determine the differences among diagonal diffusion tensor components ($D_{xx}$, $D_{yy}$, and $D_{zz}$), as well as the differences in structure coherency coefficient in porcine corneas (native and CXL) from both intact eyeballs and cornea sections, the differences in diffusion fractional anisotropy, structure coherency coefficient, and water content among three groups of rat tail tendons (healthy and thermally treated), and the differences in diffusion fractional anisotropy in scaffold experiments. A two-way ANOVA with Bonferroni post-hoc test was applied to determine the 3D diffusion differences among different experiment groups and 2D FRAP results differences in cornea sections (native and CXL) with experimental conditions and diffusion direction as the fixed factors. An independent two-sided t-test was performed to determine differences between the 2D diffusion tensor components in the cornea section experiment, differences in diffusion fractional anisotropy and cornea thickness in two groups of porcine corneas (native and CXL), and

differences in structure coherency coefficient between two groups of scaffolds (aligned and random). When multiple measurements were performed at different locations in the same sample, measurement location was taken as a covariate for the analysis. All analyses were performed in SPSS (IBM SPSS Statistics, Version 24.0, IBM Corp., Armonk, NY). Significant differences were reported at $\alpha < 0.05$, with descriptive statistics reported as mean ± standard deviation.

**Reporting summary**. Further information on research design is available in the Nature Research Reporting Summary linked to this article.

## Data availability
The raw image data that support the findings of this study are available from the corresponding author upon reasonable request. Source data are provided with this paper.

## Code availability
The code for data analysis is available from the corresponding author upon reasonable request and is limited to non-commercial use.

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

## Acknowledgements

We would like to thank Dr. Elizabeth H. Slate for her valuable assistance in statistical analysis. This work was supported by the National Institutes of Health (NIH) grants P20GM121342, R03DE018741, and R01DE021134 to H.Y., and R21GM104683 and P20GM103499 to T.Y. This work was also supported by NIH T32 post-doctoral fellowship DE017551 to R.G.H., and F32 post-doctoral fellowship DE027864 to M.C.C. This work is also supported in part by the Cell & Molecular Imaging Shared Resource, Hollings Cancer Center, Medical University of South Carolina (P30CA138313), the SC COBRE in Oxidants, Redox Balance, and Stress Signaling (P20GM103542), and the Shared Instrumentation Grant S10OD018113.

## Author contributions

The manuscript was written through contributions by all authors. All authors have given approval to the final version of the manuscript. H.Y. conceived and directed the study. H.Y., T.Y., P.C., and X.C. designed the experiments and the optical system. T.Y. and X.C. set up the optical system and wrote the system software. P.C., X.C., and R.G.H. performed the diffusion experiments. P.C. and B.J.D. fabricated the fiber-based scaffolds.

P.C. and R.G.H. conducted the SEM, SHG, and polarization imaging. P.C., S.C., and H.Y. wrote the software and analyzed the FRAP data. P.C., X.C., J.J.Y., M.C.C., M.J.K., T.Y., and H.Y. wrote the manuscript.

## Competing interests

Clemson University has submitted a patent application (application number: 63/052,726) to the U.S. Patent and Trademark Office pertaining to the method (optical system and 3D diffusion data analysis algorithm) and the applications presented in this manuscript. H.Y., T.Y., P.C., and X.C. are named inventors on this patent application. The remaining authors declare no competing interests.
