## [Peer Review File · Nature Communications]

Reviewers' Comments:

Reviewer #2:

Remarks to the Author:

General Comments:

In this manuscript, a new method for noninvasively quantifying 3D anisotropic solute diffusion tensor in biological tissues has been presented. The method is based on Fourier transform analysis of 3D images from fluorescent recovery after photobleaching (FRAP). It is claimed that this new method can be used for determining various biomolecules with diffusivities up to $51 \text{ } \mu\text{m}^2/\text{s}$. The applications of this technology to investigating anisotropic diffusion behavior of different solutes in cornea, tendon and biomimetic scaffolds were presented in this manuscript.

The claimed method is novel and its applications are new.

This paper will be of interest to many researchers in the field.

The new information on 3D anisotropic diffusion tensor will help understand transport properties of these tissues, and elucidate the relationship between tissue structure and transport properties (e.g., diffusion tensors of various solutes).

The new method was validated with different molecules and media. The information on validation was provided. The technology seems sound.

Ideally, use of materials with known 3D anisotropic diffusion tensor for further validating their method is preferred. However, such a material may not be readily available.

The manuscript was clearly written. Sufficient methodological detail for the experiments and data analysis is provided. The statistical analysis of the data is sound

However, there is one major question/concern, regarding to the calculation of fractional anisotropy (Equation in Page 25). See Specific Comments below for details. Some suggestions for minor changes are also provided below.

Specific Comments:

Page 4: line 15: "Recently, ...to reconstruct the 3D diffusion properties in...".

It is suggested to change the word "reconstruct" to "determine".

Page 6, line 5: "...sets an upper limit on diffusion measurements at...".

This sentence is not obvious to those who are unfamiliar with the FRAP method. It may be better to say "... limits the diffusion measurements for solute diffusivities at $0.5 \text{ } \mu\text{m}^2/\text{s}$ and lower", or similar sentence to avoid potential misunderstanding.

Page 11, starting at line 24: "Yet surprisingly, we found that the magnitude of 3D diffusion anisotropy was significantly lower in CXL corneas than native corneas (Fig. 4f) while the structural anisotropy showed no difference between CXL and native corneas (Fig. 4d). This result demonstrated that the diffusion (functional property) measurement is more sensitive to detect tissue structure and composition changes, compared to morphological structure measurement." This is the only major question/concern this reviewer has.

In Page 25, an equation for quantifying diffusion anisotropy, i.e., fractional anisotropy (FA), is listed. This equation is valid only when the x-y-z coordinates are in the principal directions of diffusion. See "Microstructural and Physiological Features of Tissues Elucidated by Quantitative-Diffusion-Tensor MRI", Journal of Magnetic Resonance, Series B, 111:209-219, 1996, Article No. 0086. Is it possible that there is an error in FA calculation reported in Fig. 4f? If so, the corresponding sentences in the manuscript need revisions accordingly.

Page 12, starting at line 4: "The solute diffusivity was also much slower than the value measured by LiFT-FRAP, owing to the use of viscous dextran solution for the prevention of cornea sections from tissue swelling. Additionally, although anisotropic diffusion was found in both native and CXL groups (Fig. 4g), 2D FRAP results showed that the diffusion anisotropy is slightly higher in CXL sections than native sections (Fig. 4h), which is opposite from the significantly decreased diffusion anisotropy seen in LiFT-FRAP results (Fig. 4f)."

Is it possible that the use of dextran solution causes tissue section shrinking in 2D cases? See

above comment for anisotropic characterization.

Page 13, in second paragraph: "The investigation on 3D diffusion changes lag behind mechanical studies even though it is well known that molecular diffusion is important to tissue functions"

The word "lag" should be "lags"?

It is suggested to delete the word "changes".

Page 15, starting at line 11: "To first test..., we measured..."

It is suggested to change it into "To test..., we first measured..."

Page 18, starting at 17: "First, invasive... Second, ... Finally, ...or measurements."

It seems that some similar points are repeated.

Basically, tissue dissection procedures alter mechanical and chemical environment (or loading conditions) of the tissue, causing changes in tissue composition (e.g., water content) and morphology (e.g., collagen alignment), concomitantly, resulting in different diffusivities from those in the native conditions. It is suggested to make this part of the discussion more concise.

Page 19, in second paragraph: "The discrepancies may be attributed to the excision procedures involved in the latter study."

Is it possible that the discrepancies are also caused by the different method or different solute size?

Page 19, last sentence: "Additionally, we found no changes of ..."

Please see the comment on the calculation of FA mentioned above. In addition, is it possible that the "morphological structure analysis" used in this study may be not sensitive enough to the changes?

Page 20, in second paragraph: "Many studies have ..., yet very few have been able to show the mechanistic link between pathological structure and function."

This sentence is not clear. While transport properties (such as diffusion coefficient) correlate with tissue structure, the measurement of diffusivities is an indirect way for characterizing tissue structure.

Page 20, last paragraph: "By elucidating the link between tissue structure and function via accurate quantification of 3D diffusion behavior in scaffolds, ..."

Different tissues have different functions. For example, the main function of cornea is light transmission. "Accurate quantification of 3D diffusion behavior" may provide additional information on tissue structure indirectly, and transport properties directly, but not on tissue function. Please modify the sentence to avoid misleading.

Page 21, line 6: It is suggested to change the word "clear" to "transparent".

Page 25, the equation for calculating FA: See the relevant comment above.

Page 36, in References

It is suggested to carefully check the spelling of authors' name and make the format consistent.

Page 41, in Caption for Fig. 1: The meaning of "molecular dynamics" is not clear. It is suggested to delete it. In addition, it is suggested to mention that the simulation is for illustration purpose, not related to actual FRAP experiments presented in this manuscript.

Page 44, in Caption for Fig. 3a: "The volume rate of ... is 125 ms".

It is suggested to change it into "The rate of image acquisition is 8 volume images per second", or similar sentence.

Page 46, in Fig. 4e, Fig. 4g, and Fig. 4i

Please use D'_{xx} in order to be consistent with those in Supplementary Notes (p. 23).

Page 48, Caption for Fig. 5

The authors should include the zero-flux boundary condition here.

Page 49, line 2: "...with the diffusion input measured..."

It is suggested to change it into "...with the diffusivities measured..."

Please delete extra ") " in line 4.

Reviewer #3:

Remarks to the Author:

Manuscript Review

Title: "LiFT-FRAP: A noninvasive fluorescence imaging-based platform measures 3D anisotropic extracellular diffusion"

Corresponding Author: Hai Yao

Summary: This manuscript describes a method for measuring the 3D diffusion tensor by integrating rapid light-sheet volumetric imaging with spatial frequency analysis. The technique is optimized for noninvasive measurements in biological tissues and is applied to resolve questions about anisotropy in molecular diffusion in the cornea, tendon tissues, and biomimetic scaffolds. The authors successfully demonstrated statistically significant differences in diffusivity depending on the alignment within biological tissues through recovery of the diffusion tensor. It is recommended for publication following minor revisions.

Comments:

1. The authors indicate as an enabling innovation the ability to record fast diffusion through acquisition of volumes at frame rates of 8 volumes / second, which presumably allows access to recoveries requiring a few seconds or less that are inaccessible through prior established methods. However, this fast acquisition speed appears to be at odds with the 1s exposures used to perform the photobleaching. If diffusion arises fast enough for the 8 volumes/s speeds to be advantageous, isn't significant diffusion already arising during the timescale of the photobleaching? Can the authors include a description of the role diffusion during photobleaching may play in the interpretation of their final results?
2. The photobleaching is performed through multi-photon excitation by focusing 200-700 mW of laser power to a diffraction-limited spot through a 0.8 NA 40x objective for durations of 1s. This is a remarkably high laser power density, which is well in excess of commonly accepted damage thresholds in multi-photon microscopy. Can the authors provide evidence that local heating associated with this power density of incident light is not influencing the sample and/or the subsequent diffusivity? A more complete description of control experiments to assess the role of local heating and sample damage would be beneficial.
3. It appears that the introduction section is lacking a discussion of other 3D FRAP techniques (e.g. DOI: 10.1117/12.2539683) and attempts to correct for the biases introduced in 2D-FRAP measurements through alternative diffusion models (e.g. DOI: 10.1016/S0006-3495(03)74649-9). It is recommended that the authors provide more context for where their technique stands in relation to other FRAP techniques.
4. The more complex samples considered in this study likely exhibit a degree of anomalous diffusion. It does not appear that the diffusion model used in LiFT-FRAP accounts for anomalous diffusion. The authors are encouraged to provide a justification for assuming normal diffusion in these experiments.
5. 2) Why is the superoinferior diffusion (D_{yy}) equivalent to the mediolateral diffusion (D_{xx}) and not the anteroposterior diffusion (D_{zz})? For fibrous collagen structures, one would naively expect to be unique when aligned along the fiber axis (x' -axis), relative to the two equivalent directions

orthogonal to the fiber axis. However, in Figure 4, diffusion along x' and y' is statistically similar, while z' is unique. The authors are encouraged to provide additional explanation for these observations.

Re: NCOMMS-20-36943

LiFT-FRAP: A noninvasive fluorescence imaging-based platform measures 3D anisotropic extracellular diffusion

The initial reviewer #1 dropped out so our reviewers are #2 and #3.

Response to Reviewer #2:

General Comments:

In this manuscript, a new method for noninvasively quantifying 3D anisotropic solute diffusion tensor in biological tissues has been presented. The method is based on Fourier transform analysis of 3D images from fluorescent recovery after photobleaching (FRAP). It is claimed that this new method can be used for determining various biomolecules with diffusivities up to $51 \mu\text{m}^2/\text{s}$. The applications of this technology to investigating anisotropic diffusion behavior of different solutes in cornea, tendon and biomimetic scaffolds were presented in this manuscript. The claimed method is novel and its applications are new.

This paper will be of interest to many researchers in the field.

The new information on 3D anisotropic diffusion tensor will help understand transport properties of these tissues, and elucidate the relationship between tissue structure and transport properties (e.g., diffusion tensors of various solutes).

The new method was validated with different molecules and media. The information on validation was provided. The technology seems sound.

Ideally, use of materials with known 3D anisotropic diffusion tensor for further validating their method is preferred. However, such a material may not be readily available.

The manuscript was clearly written. Sufficient methodological detail for the experiments and data analysis is provided. The statistical analysis of the data is sound.

However, there is one major question/concern, regarding to the calculation of fractional anisotropy (Equation in Page 25). See Specific Comments below for details. Some suggestions for minor changes are also provided below.

Response: We thank the reviewer for noting the novelty and the significance of our work. We also appreciate the reviewer's suggestions and comments to help improve the quality of our manuscript. We agree with the reviewer that using a material with known 3D anisotropic diffusion properties for further validation would be ideal. However, as the reviewer suggested, standardized materials with known 3D anisotropic diffusion tensors are not readily available, owing to the lack of tools for generating these measurements. Addressing this gap was the main driving force in our development of LiFT-FRAP. We did our best to validate our method with a series of materials using manipulations with known effects. We manipulated the tissue structure

and composition properties to demonstrate the sensitivity of our method. We also fabricated both isotropic and anisotropic scaffolds to prove that our method can perform noninvasive 3D diffusion tensor measurements in biological systems. We will continue to validate our method in future studies, and we will test our method on standard materials once they become available.

We thank the reviewer for the question on the calculation of fractional anisotropy. As suggested by the reviewer, we have recalculated the fractional anisotropy using the principal diffusivities of the 3D diffusion tensor as defined in the Magnetic Resonance Imaging (MRI) field. We have updated all related figures in the revised manuscript. Compared to the previous calculation, there were only minor changes in fractional anisotropy; thus, all conclusions remain the same. The similarity of the results is expected as the sample coordinates in our work were roughly aligned to the principal diffusion directions. The off-diagonal components of the reported 3D diffusion tensors are much smaller than the diagonal components, approaching zero.

Along with the above points, we have responded to each of the specific comments below.

Specific Comments:

Page 4: line 15: “Recently, ...to reconstruct the 3D diffusion properties in...”.
It is suggested to change the word “reconstruct” to “determine”.

Response: As suggested, we have changed the word in the revised manuscript.

Page 6, line 5: “...sets an upper limit on diffusion measurements at...”.

This sentence is not obvious to those who are unfamiliar with the FRAP method. It may be better to say “... limits the diffusion measurements for solute diffusivities at $0.5 \text{ } \mu\text{m}^2/\text{s}$ and lower”, or similar sentence to avoid potential misunderstanding.

Response: We thank the reviewer for pointing out potential misunderstanding. As suggested, we have made changes in the revised manuscript to enhance the accessibility of our work to the general audience.

Page 11, starting at line 24: “Yet surprisingly, we found that the magnitude of 3D diffusion anisotropy was significantly lower in CXL corneas than native corneas (Fig. 4f) while the structural anisotropy showed no difference between CXL and native corneas (Fig. 4d). This result demonstrated that the diffusion (functional property) measurement is more sensitive to detect tissue structure and composition changes, compared to morphological structure measurement.”

This is the only major question/concern this reviewer has.

In Page 25, an equation for quantifying diffusion anisotropy, i.e., fractional anisotropy (FA), is listed. This equation is valid only when the x-y-z coordinates are in the principal directions of diffusion. See “Microstructural and Physiological Features of Tissues Elucidated

by Quantitative-Diffusion-Tensor MRI”, Journal of Magnetic Resonance, Series B, 111:209-219, 1996, Article No. 0086. Is it possible that there is an error in FA calculation reported in Fig. 4f ? If so, the corresponding sentences in the manuscript need revisions accordingly.

Response: We thank the reviewer for the question. As mentioned by the reviewer, the fractional anisotropy (FA) calculation as delineated in the MRI study is based on the principal values of the 3D diffusion tensor. To make our results consistent with the literature report, we calculated the principal diffusivities of the 3D diffusion tensor and recalculated the FA value. Figures 4f, 4h, and 4j, as well as the equation and related description in the Method section (Line 527 – 530 in Page 25), have been updated in the revised manuscript. The new FA results were found to be similar to our previous FA results, as the sample coordinates used in our study are roughly aligned with the principal diffusion directions, so the FA in CXL corneas is still significantly lower than the FA in native corneas.

Page 12, starting at line 4: “The solute diffusivity was also much slower than the value measured by LiFT-FRAP, owing to the use of viscous dextran solution for the prevention of cornea sections from tissue swelling. Additionally, although anisotropic diffusion was found in both native and CXL groups (Fig. 4g), 2D FRAP results showed that the diffusion anisotropy is slightly higher in CXL sections than native sections (Fig. 4h), which is opposite from the significantly decreased diffusion anisotropy seen in LiFT-FRAP results (Fig. 4f).”

Is it possible that the use of dextran solution causes tissue section shrinking in 2D cases? See above comment for anisotropic characterization.

Response: We thank the reviewer for the question. In this study, we added dextran to our incubation solution to help maintain cornea hydration, as cornea sections are shown to swell significantly in normal saline solution. 20% dextran solution is used in the protocol of the FDA approved CXL treatment ¹. However, it has been reported that 20% dextran solution could dehydrate the cornea, leading to cornea thinning ². Therefore, in our work, we intentionally decreased the dextran concentration to 17.5% for the staining. With the 17.5% dextran solution, we found no visually observable changes in cornea section hydration during the staining process. However, as mentioned by the reviewer, it is possible that this lower concentration dextran solution may still cause tissue section shrinking, and the shrinkage may partially explain the decrease in solute diffusivity measured using 2D FRAP. Such difficulty in handling tissue sections for 2D measurements demonstrates the need for a platform like our LiFT-FRAP system for 3D noninvasive diffusion measurement. We have added “partially” in this sentence (Line 249 in Page 12) to avoid making the claim that viscous solution infiltration was the only reason for the diffusion decrease. For the characterization of diffusion anisotropy, we have recalculated the fractional anisotropy with the principal diffusivities as suggested by the reviewer. Figure 4f, 4h, and 4j have been updated in the revised manuscript. The new results are similar to our previous calculations, and the conclusion remains the same.

Page 13, in second paragraph: “The investigation on 3D diffusion changes lag behind mechanical studies even though it is well known that molecular diffusion is important to tissue functions”
The word “lag” should be “lags”?
It is suggested to delete the word “changes”.

Response: We apologize for the typo and the phrase has been corrected in the revised manuscript.

Page 15, starting at line 11: “To first test..., we measured...”
It is suggested to change it into “To test..., we first measured...”

Response: As suggested, we have changed the phrase in the revised manuscript.

Page 18, starting at 17: “First, invasive.... Second, ... Finally, ...or measurements.”
It seems that some similar points are repeated.
Basically, tissue dissection procedures alter mechanical and chemical environment (or loading conditions) of the tissue, causing changes in tissue composition (e.g., water content) and morphology (e.g., collagen alignment), concomitantly, resulting in different diffusivities from those in the native conditions. It is suggested to make this part of the discussion more concise.

Response: We thank the reviewer for the very helpful suggestion. We have reworked this section to make this part of discussion more concise in the revised manuscript (Line 322 – 396 in Page 18).

Page 19, in second paragraph: “The discrepancies may be attributed to the excision procedures involved in the latter study.”
Is it possible that the discrepancies are also caused by the different method or different solute size?

Response: We agree with the reviewer that beyond the excision procedures, the discrepancies could be attributable to the different methods or solute sizes used. However, we believe that the difference in sample preparation procedures is the main reason for the discrepancies. In the fluorescence correlation spectroscopy (FCS) study, isotropic diffusion was reported with probes ranging from 3 kDa to 2,000 kDa in molecular weight. In the MRI-based diffusion tensor imaging (DTI) study, the 3D diffusion of water molecules (18 Da) was found to be anisotropic. However, based on modeling analysis³ and our results in this paper, larger molecules should display higher diffusion anisotropy. Therefore, given that anisotropic diffusion with water molecules was detected in the DTI study, we should expect more significant anisotropic diffusion with the larger molecules used in the FCS study. This led us to believe that the different sample preparation procedure used by the two different methods might be the major cause of the discrepancies. In the FCS study, the corneas were dissected, while in the DTI study, intact corneas were used. Our results in Fig 4c, 4d demonstrate the dramatic change in cornea structure after dissection, which can significantly alter tissue diffusion properties.

Page 19, last sentence: “Additionally, we found no changes of ...”

Please see the comment on the calculation of FA mentioned above. In addition, is it possible that the “morphological structure analysis” used in this study may be not sensitive enough to the changes?

Response: As suggested by the reviewer, we have recalculated the fractional anisotropy (FA) with the principal diffusivities, and the new results have been updated in Figure 4f. The results are similar to our previous calculations, with significantly higher FA in native corneas than CXL corneas. We agree that the morphological structure analysis used in this study might not be sensitive enough to detect the structural changes caused by CXL. The morphological analysis used in this work mainly characterized the alignment of the tissue fibers, which only reflects one aspect of the tissue structure properties. To further investigate 3D tissue structure changes, we also measured the thickness of the corneas to find that native corneas were significantly thicker than CXL corneas (see the results in Supplementary Fig. 6e). These results indirectly showed that cornea lamellae were more compact in CXL corneas than in native corneas, which could affect the molecular diffusion properties. Such changes in thickness can be regarded as another dimension of the 3D tissue structure changes. In the revised manuscript, we have clarified that our morphological analysis is only based on fiber alignment and included the discussion on the cornea thickness measurements in Page 19 - 20 (Line 415 – 420). We have also revised our assessment of the sensitivity of our 3D diffusion measurement to detect overall 3D tissue structure changes. The sentence “This result demonstrated ... morphological structure measurement” in the Result section in Page 11 has been removed.

Page 20, in second paragraph: “Many studies have ..., yet very few have been able to show the mechanistic link between pathological structure and function.”

This sentence is not clear. While transport properties (such as diffusion coefficient) correlate with tissue structure, the measurement of diffusivities is an indirect way for characterizing tissue structure.

Response: We thank the reviewer for pointing this out. We agree that the sentence is not clearly delivered. We also agree with the reviewer that the measurement of molecular diffusion is an indirect way to characterize tissue structure. With this sentence, we are attempting to draw the connection between diffusion measurement with the tissue biochemical environment. As shown in our simulation work, we can estimate the molecular distribution within tissues using quantitative diffusion information, thereby providing quantitative evidence towards elucidating the effects of pathological changes on molecular mechanisms. In the revised manuscript, we have changed the phrase as suggested and included more context to clarify our point on the connection between diffusion and the biochemical environment (Line 430 – 434 in Page 20).

Page 20, last paragraph: “By elucidating the link between tissue structure and function via accurate quantification of 3D diffusion behavior in scaffolds, ...”

Different tissues have different functions. For example, the main function of cornea is light

transmission. “Accurate quantification of 3D diffusion behavior” may provide additional information on tissue structure indirectly, and transport properties directly, but not on tissue function. Please modify the sentence to avoid misleading.

Response: We apologize for the misleading sentence. In the revised manuscript, we have deleted “elucidating the link between tissue structure and function via,” and mainly discussed the connection between diffusion and scaffold development instead of directly relating diffusion behavior to tissue function.

Page 21, line 6: It is suggested to change the word “clear” to “transparent”.

Response: As suggested, we have changed the word in the revised manuscript.

Page 25, the equation for calculating FA: See the relevant comment above.

Response: As suggested by the reviewer, we have changed this equation with the principal diffusivities in the revised manuscript. All the FA results have been recalculated and all related figures have been updated in the revised manuscript.

Page 36, in References

It is suggested to carefully check the spelling of authors’ name and make the format consistent.

Response: We appreciate the reviewer’s suggestions. We apologize for the inconsistency in formatting. We have made the appropriate corrections where needed, and we have carefully checked the spelling and formatting in the revised manuscript.

Page 41, in Caption for Fig. 1: The meaning of “molecular dynamics” is not clear. It is suggested to delete it. In addition, it is suggested to mention that the simulation is for illustration purpose, not related to actual FRAP experiments presented in this manuscript.

Response: We thank the reviewer for the very insightful suggestions. We have made changes as suggested in the revised manuscript.

Page 44, in Caption for Fig. 3a: “The volume rate of ... is 125 ms”.

It is suggested to change it into “The rate of image acquisition is 8 volume images per second”, or similar sentence.

Response: As suggested, we have made the changes in the revised manuscript.

Page 46, in Fig. 4e, Fig. 4g, and Fig. 4i

Please use D'_{xx} in order to be consistent with those in Supplementary Notes (p. 23).

Response: We apologize for the typo and a correction has been made.

Page 48, Caption for Fig. 5

The authors should include the zero-flux boundary condition here.

Response: We appreciate the reviewer's helpful comments. We have added the boundary condition in the caption of the revised manuscript (Line 1007 – 1008 in Page 49).

Page 49, line 2: "...with the diffusion input measured..."

It is suggested to change it into "...with the diffusivities measured..."

Please delete extra ") " in line 4.

Response: We thank the reviewer for the suggestions. The phrase has been changed as suggested. We also apologize for the typo.

Response to Reviewer #3:

Summary: This manuscript describes a method for measuring the 3D diffusion tensor by integrating rapid light-sheet volumetric imaging with spatial frequency analysis. The technique is optimized for noninvasive measurements in biological tissues and is applied to resolve questions about anisotropy in molecular diffusion in the cornea, tendon tissues, and biomimetic scaffolds. The authors successfully demonstrated statistically significant differences in diffusivity depending on the alignment within biological tissues through recovery of the diffusion tensor. It is recommended for publication following minor revisions.

Response: We thank the reviewer for their well-articulated summary of our work and positive feedback. We have provided our responses to each of the reviewer's comments below.

Comments:

1. The authors indicate as an enabling innovation the ability to record fast diffusion through acquisition of volumes at frame rates of 8 volumes / second, which presumably allows access to recoveries requiring a few seconds or less that are inaccessible through prior established methods. However, this fast acquisition speed appears to be at odds with the 1s exposures used to perform the photobleaching. If diffusion arises fast enough for the 8 volumes/s speeds to be advantageous, isn't significant diffusion already arising during the timescale of the photobleaching? Can the authors include a description of the role diffusion during photobleaching may play in the interpretation of their final results?

Response: We thank the reviewer for the question. Based on our previously published theory, any diffusion during the bleaching process does not affect our diffusion measurement. This is one of the major advantages of the frequency domain-based spatial Fourier analysis (SFA) method used in our work⁴, compared to the conventional space domain-based methods⁵⁻⁸. With our method, the initial condition of bleaching region is very flexible. The bleaching spot can be

any shape, and diffusion during bleaching is allowed. These features greatly enhance the accuracy and convenience of FRAP measurement. The incorporation of the fast 3D imaging approach allows us to collect enough data points for curve-fitting even when the bleaching region has partially recovered due to diffusion during the bleaching process, therefore enabling fast diffusion measurement. We have included the discussion of this advantage of LiFT-FRAP in the revised manuscript (Line 399 – 401 in Page 19).

2. The photobleaching is performed through multi-photon excitation by focusing 200-700 mW of laser power to a diffraction-limited spot through a 0.8 NA 40x objective for durations of 1s. This is a remarkably high laser power density, which is well in excess of commonly accepted damage thresholds in multi-photon microscopy. Can the authors provide evidence that local heating associated with this power density of incident light is not influencing the sample and/or the subsequent diffusivity? A more complete description of control experiments to assess the role of local heating and sample damage would be beneficial.

Response: We thank the reviewer for raising this important question regarding possible heating or photodamage in the bleaching process. These effects are always the principal concerns in any FRAP experiment, as high intensity laser power is normally needed for photobleaching. Therefore, we were very cautious when we performed the experiments. Here, we first want to clarify that the mentioned laser power (200-700 mW) was measured at the shutter location as indicated in Supplementary Table 4 of our initial manuscript. Due to attenuation at each optics along the optical pathway, the actual laser power out of the bleaching objective ends up being only 40% of that measured at the shutter. This would result in a range of 80-280 mW used for bleaching samples. This level of laser power is well within the power range reported in the literature. For instance, 500 mW of laser power has been used for photobleaching in two-photon based FRAP experiments⁹. It has also been reported that the use of over 300 mW of laser power at 736 nm shows no evidence of photodamage in brain slides¹⁰.

To monitor the potential tissue damage caused by the bleaching laser, we built a second harmonic generation (SHG) imaging unit on our light-sheet system. SHG images captured any collagen structural changes after photobleaching. We imaged the rat tail tendon structures before and after the bleaching process. The bleaching laser power was set as 400 mW (at the shutter location, so the actual laser power right after the bleaching objective was 160 mW), the maximum laser power used for tendon experiments. The result showed that the SHG images before and after bleaching at the same location was identical (see images below). No structural or SHG signal changes were observed. Likewise, in our cornea measurements, due to the weak backward SHG signal of cornea tissue and the 45° alignment of our detection objective, cornea structure was unable to be imaged with our current setup. However, from the 3D postbleaching images (Supplementary Figure 6c and 6d), no visual photodamage was observed at the bleaching region, which indirectly demonstrated that the bleaching laser power did not affect the sample.

SHG images of rat tail tendon before and after the bleaching process. Scale bar, 20 μm .

As suggested by the reviewer, we also examined the heating effect of bleaching laser power on the samples. It has been reported that the major thermal effect of two-photon imaging comes from the one-photon absorption ¹¹. The maximum temperature rise at the laser focal point can be estimated with the following equation ¹¹:

$$\Delta T = \frac{\mu_a E f_p}{4\pi k_t} \ln \left(1 + \frac{2t_{pix}}{\tau_c} \right) + \frac{\mu_a E}{2\pi k_t \tau_c}$$

where μ_a is the absorption coefficient ($\sim 0.03 \text{ cm}^{-1}$ at 800 nm for the collagen ¹²), E is the energy of a single laser pulse, f_p is the laser repetition frequency (80 MHz), k_t is the water thermal conductivity ($\sim 0.6 \text{ WK}^{-1}\text{m}^{-1}$), t_{pix} is the pixel dwell time during bleaching scan (90 μs), and τ_c is the thermal time constant (70 ns). For the maximum laser power used in this study (280 mW), E is 3.5 nJ. From this, we estimate a temperature increase of only 0.9 K. This result is expected as the tissues we tested have no pigments or other strong near-infrared absorbers. This result is also comparable with literature results. For example, it has been reported that the local temperature increase with irradiation of an 800 nm laser at a power of 100 mW is $\sim 0.2 \text{ K}$ ¹³. In brain tissue two-photon imaging, it has also been shown that the maximum heating response is $\sim 1.8 \text{ K}/100 \text{ mW}$ ¹⁴. Such a small temperature increase should not significantly change the solute diffusion properties nor cause tissue damage.

We have adjusted the laser power values in Supplementary Table 4 to reflect the laser power measured right after the objective instead at the shutter to keep our report consistent with the literature in the revised manuscript. We have also clarified that no photodamage of tissue

samples was observed in any experiments in the Method section of the revised manuscript in Page 24 – 25 (Line 515 – 517).

3. It appears that the introduction section is lacking a discussion of other 3D FRAP techniques (e.g. DOI: 10.1117/12.2539683) and attempts to correct for the biases introduced in 2D-FRAP measurements through alternative diffusion models (e.g. DOI: 10.1016/S0006-3495(03)74649-9). It is recommended that the authors provide more context for where their technique stands in relation to other FRAP techniques.

Response: We thank the reviewer for the insightful recommendations. We recognize that there are other powerful 3D FRAP techniques for diffusion measurement. These methods, however, mainly assume isotropic diffusion and are unable to generate a 3D anisotropic diffusion tensor. We agree with the reviewer's comments to include more context about other related 3D FRAP techniques to demonstrate the novelty of our technique to the field. Towards this end, we have expanded the introduction section with more information on other related 3D FRAP techniques and added the relevant citations in Page 5 – 6 (Line 101 – 111) of the revised manuscript.

4. The more complex samples considered in this study likely exhibit a degree of anomalous diffusion. It does not appear that the diffusion model used in LiFT-FRAP accounts for anomalous diffusion. The authors are encouraged to provide a justification for assuming normal diffusion in these experiments.

Response: We thank the reviewer for this question. The tissue samples used in this work have well-organized collagen fiber structure. From the SHG images of both biological tissues, we can see that the tissue structure is uniform, especially at the scale of our observation volume, $76 \times 76 \times 76 \mu\text{m}^3$. Therefore, it is reasonable to assume that the 3D diffusion tensor is constant within our imaging volume, and that the diffusion we measured in this work is the effective diffusion. Additionally, we used inert standard probes for all of the experiments, so there should be no transient binding effect, which is one of the major contributors to the anomalous diffusion phenomenon. Moreover, it has been reported that SFA allows anomalous diffusion analysis, as it can perform simultaneous measurement of both normal diffusion and binding rate in 2D^{9,15}. Since our technique can record the 3D fluorescence recovery process due to normal and/or anomalous diffusion, it would be interesting to investigate the ability of our method to quantify more complicated diffusion phenomena in a 3D space in our future work. We have included the discussion on this potential future study in Page 21 (Line 456 – 460) of the revised manuscript.

5. 2) Why is the superoinferior diffusion (D_{yy}) equivalent to the mediolateral diffusion (D_{xx}) and not the anteroposterior diffusion (D_{zz})? For fibrous collagen structures, one would naively expect to be unique when aligned along the fiber axis (x' -axis), relative to the two equivalent directions orthogonal to the fiber axis. However, in Figure 4, diffusion along x' and y' is statistically similar, while z' is unique. The authors are encouraged to provide additional explanation for these observations.

Response: We thank the reviewer for the question. The interesting 3D diffusion results for the cornea tissue is related to its unique fiber structure. Cornea tissue exhibits the lamellae collagen fiber structure as shown in Figure. 4c and Supplementary Figure. 6b, and the direction of the fibers within adjacent cornea lamellae are different, mostly perpendicular to each other¹⁶. As each lamellae layer is very thin, we can regard the overall fiber alignment in the transverse plane ($x'y'$ plane) as random within our 3D imaging volume. Therefore, we should not expect the diffusion behavior in cornea tissue to be comparable to that in tendon tissue. The diffusion along the anteriorposterior direction (D'_{zz}) is perpendicular to the fiber axis; therefore, the D'_{zz} has the smallest value. Meanwhile, the molecular diffusion along the mediolateral (D'_{xx}) and superoinferior (D'_{yy}) directions are parallel to the lamellae layer. They experience similar impedance within the 3D imaging volume; therefore, the D'_{xx} and D'_{yy} are equivalent. We have included additional explanation in the corresponding Result section in Page 11 (Line 225 – 231).

Reference

- 1 Wollensak, G., Spoerl, E. & Seiler, T. Riboflavin/ultraviolet-a-induced collagen crosslinking for the treatment of keratoconus. *Am J Ophthalmol* **135**, 620-627 (2003).
- 2 Oltulu, R. *et al.* Intraoperative corneal thickness monitoring during corneal collagen cross-linking with isotonic riboflavin solution with and without dextran. *Cornea* **33**, 1164-1167 (2014).
- 3 Stylianopoulos, T., Diop-Frimpong, B., Munn, L. L. & Jain, R. K. Diffusion anisotropy in collagen gels and tumors: the effect of fiber network orientation. *Biophys J* **99**, 3119-3128 (2010).
- 4 Shi, C., Cisewski, S. E., Bell, P. D. & Yao, H. Measurement of three-dimensional anisotropic diffusion by multiphoton fluorescence recovery after photobleaching. *Ann Biomed Eng* **42**, 555-565 (2014).
- 5 Axelrod, D., Koppel, D. E., Schlessinger, J., Elson, E. & Webb, W. W. Mobility measurement by analysis of fluorescence photobleaching recovery kinetics. *Biophys J* **16**, 1055-1069 (1976).
- 6 Mazza, D. *et al.* A new FRAP/FRAPa method for three-dimensional diffusion measurements based on multiphoton excitation microscopy. *Biophys J* **95**, 3457-3469 (2008).
- 7 Braeckmans, K., Peeters, L., Sanders, N. N., De Smedt, S. C. & Demeester, J. Three-dimensional fluorescence recovery after photobleaching with the confocal scanning laser microscope. *Biophys J* **85**, 2240-2252 (2003).
- 8 Brown, E. B., Wu, E. S., Zipfel, W. & Webb, W. W. Measurement of molecular diffusion in solution by multiphoton fluorescence photobleaching recovery. *Biophys J* **77**, 2837-2849 (1999).
- 9 Geiger, A. C. *et al.* Anomalous diffusion characterization by Fourier transform-FRAP with patterned illumination. *Biophys J* **119**, 737-748 (2020).
- 10 Ding, J. B., Takasaki, K. T. & Sabatini, B. L. Supraresolution imaging in brain slices using stimulated-emission depletion two-photon laser scanning microscopy. *Neuron* **63**, 429-437 (2009).
- 11 So, P. T. C. Two-photon fluorescence microscopy: a new tool for tissue imaging and spectroscopy. *J Histotechnol* **23**, 221-228 (2000).
- 12 Sekar, S. K. *et al.* Diffuse optical characterization of collagen absorption from 500 to 1700 nm. *J Biomed Opt* **22**, 15006 (2017).
- 13 Schönle, A. & Hell, S. W. Heating by absorption in the focus of an objective lens. *Opt Lett* **23**, 325-327 (1998).
- 14 Podgorski, K. & Ranganathan, G. Brain heating induced by near-infrared lasers during multiphoton microscopy. *J Neurophysiol* **116**, 1012-1023 (2016).
- 15 Travascio, F. & Gu, W. Y. Simultaneous measurement of anisotropic solute diffusivity and binding reaction rates in biological tissues by FRAP. *Ann Biomed Eng* **39**, 53-65 (2011).
- 16 Komai, Y. & Ushiki, T. The three-dimensional organization of collagen fibrils in the human cornea and sclera. *Invest Ophthalmol Vis Sci* **32**, 2244-2258 (1991).

Reviewers' Comments:

Reviewer #2:

Remarks to the Author:

General Comments:

In this revised manuscript, the authors have adequately addressed the questions and concerns in my initial review.

No further questions.

Reviewer #3:

Remarks to the Author:

The revisions to the manuscript adequately address the majority of the comments raised in the prior submissions. The manuscript is recommended for publication following minor revisions to provide additional clarification and/or experiment evidence supporting the assertions.

The authors included in their response estimations of the temperature change within the photobleached region and concluded from those calculations that the temperature-induced perturbation to diffusion could be neglected. They further justified this assumption by references to previous studies also employing high laser powers during multi-photon induced photobleaching. Regarding the calculations, two questions arise upon review:

1) The origin of the 9 us dwell time for a single pixel was not clearly derived. From a back-of-the-envelope calculation, the $10\mu\text{m} \times 10\mu\text{m} \times 10\mu\text{m}$ bleached region contains ~ 5200 voxels of size $1.2\mu\text{m} \times 0.4\mu\text{m} \times 0.4\mu\text{m}$ dimension, corresponding to a dwell time of $\sim 200\text{us}$ per voxel for a 1s bleach.

2) The equation used in estimating the temperature change corresponds to a single focal volume with heat dissipated into an infinite adjacent bath. However, in practice, each of the illuminated voxels was immediately adjacent to other voxels also illuminated by the photobleach laser, such that the estimate above corresponds to a lower limit on the estimated change in temperature. The most relevant citation describing previous photobleaching by multiphoton excitation at high power is arguably the work done by Geiger and coworker, using 500 mW of power for photobleaching. However, that previous work used a resonant scan mirror corresponding to dwell times of ~ 50 ns per pixel followed by ~ 125 us relaxation before the next 50 ns illumination, with the total power distributed over $\sim 16,000$ pixels spanning the entire field of view. Given the fairly fast diffusivity of water ($143000 \text{ nm}^2/\text{us}$), the 125 us dissipation time was sufficient to enable recovery before the next illumination event. It is not immediately clear whether those same expectations may hold in the present study employing slower beam scanning and illuminating a 3D block of adjacent voxels.

The authors are encouraged to investigate the influence of local heating more directly by probing the recovered diffusion tensor as a function of photobleach power. Reduction in the photobleach depth should only influence signal to noise, but not the recovered diffusion tensor values if local heating is not perturbative.

Re: NCOMMS-20-36943A

LiFT-FRAP: A noninvasive fluorescence imaging-based platform measures 3D anisotropic extracellular diffusion

Response to Reviewer #2:

General Comments:

In this revised manuscript, the authors have adequately addressed the questions and concerns in my initial review.

No further questions.

Response: We thank the reviewer for their time and their acknowledgement of our efforts in addressing their previous comments.

Response to Reviewer #3:

The revisions to the manuscript adequately address the majority of the comments raised in the prior submissions. The manuscript is recommended for publication following minor revisions to provide additional clarification and/or experiment evidence supporting the assertions.

Response: We thank the reviewer for their critiques and suggestions. As requested, we have provided additional details and clarifications, as well as new experimental results to support the statements in the manuscript.

The authors included in their response estimations of the temperature change within the photobleached region and concluded from those calculations that the temperature-induced perturbation to diffusion could be neglected. They further justified this assumption by references to previous studies also employing high laser powers during multi-photon induced photobleaching.

Regarding the calculations, two questions arise upon review:

1) The origin of the 9 us dwell time for a single pixel was not clearly derived. From a back-of-the-envelope calculation, the $10\mu\text{m} \times 10\mu\text{m} \times 10\mu\text{m}$ bleached region contains ~ 5200 voxels of size $1.2\mu\text{m} \times 0.4\mu\text{m} \times 0.4\mu\text{m}$ dimension, corresponding to a dwell time of $\sim 200\mu\text{s}$ per voxel for a 1s bleach.

Response: We thank the reviewer for their question regarding the dwell time calculation. We also appreciate the estimation provided by the reviewer. We first would like to note that our

estimated dwell time is 90 μs in our previous response, instead of 9 μs . As the reviewer suggested, the dwell time for a single voxel can be estimated by calculating the total number of voxels (~ 5200) and the total time needed for a single scan of the whole volume (125 ms). In our photobleaching protocol, the bleaching volume was scanned for 8 times during the 1s bleaching process as the piezo for the z-axial scanning was driven by a sinusoidal waveform with a frequency of 4 Hz (details can be found in the Supplementary Information of our manuscript). Therefore, the total time needed to complete a single scanning of the bleaching volume is 125 ms instead of 1 s. Accordingly, the dwell time for a single voxel is $\sim 25 \mu\text{s}$. In our calculation, we intentionally overestimated the voxel dwell time to investigate the possible higher end of the local thermal effect. As shown in our previous estimation, although a longer dwell time ($\sim 90 \mu\text{s}$) was used for the temperature calculation, the estimated temperature rise ($\sim 0.9 \text{ K}$) was very small. Thus, we think there are very minor changes of temperature within the photobleached region.

2) The equation used in estimating the temperature change corresponds to a single focal volume with heat dissipated into an infinite adjacent bath. However, in practice, each of the illuminated voxels was immediately adjacent to other voxels also illuminated by the photobleach laser, such that the estimate above corresponds to a lower limit on the estimated change in temperature.

The most relevant citation describing previous photobleaching by multiphoton excitation at high power is arguably the work done by Geiger and coworker, using 500 mW of power for photobleaching. However, that previous work used a resonant scan mirror corresponding to dwell times of $\sim 50 \text{ ns}$ per pixel followed by $\sim 125 \mu\text{s}$ relaxation before the next 50 ns illumination, with the total power distributed over $\sim 16,000$ pixels spanning the entire field of view. Given the fairly fast diffusivity of water ($143000 \text{ nm}^2/\text{us}$), the 125 μs dissipation time was sufficient to enable recovery before the next illumination event. It is not immediately clear whether those same expectations may hold in the present study employing slower beam scanning and illuminating a 3D block of adjacent voxels.

The authors are encouraged to investigate the influence of local heating more directly by probing the recovered diffusion tensor as a function of photobleach power. Reduction in the photobleach depth should only influence signal to noise, but not the recovered diffusion tensor values if local heating is not perturbative.

Response: We thank the reviewer for their comments and suggestions. We agree with the reviewer that the actual change of temperature might be higher than what our model estimated as the bleaching voxels were next to each other in our bleaching protocol. We also thank the reviewer for pointing out the differences between our bleaching protocol and the one (Geiger and coworker) in the literature. We agree with the reviewer that it could be hard to directly compare our work with the literature study due to the protocol differences. As suggested by the reviewer, we performed carefully controlled LiFT-FRAP experiments in glycerol solutions using different bleaching laser powers: 160 mW, 200 mW, and 240 mW (laser power measured right after the objective). These bleaching laser powers cover the power range used in our manuscript. The

figure shown below is the 3D diffusion tensor results of sodium fluorescein in 80% glycerol solution from LiFT-FRAP experiments with different bleaching laser power. The results showed that there were no significant diffusion differences among the three groups ($p = 0.7$, one-way ANOVA). The molecular diffusivities did not change with the alteration of bleaching laser power, which suggested that the influence of local heating on molecular diffusion is negligible. We have added the new experimental data shown below as Supplementary Figure 10 in the revised manuscript. We have also clarified that the influence of local heating caused by the bleaching laser illumination (within the laser power range in our studies) on the molecular diffusion measurement is negligible in the Method section of the revised manuscript in Page 25.

(a) 3D diffusion tensors of sodium fluorescein in 80% glycerol solution with different bleaching laser powers: 160 mW ($n=13$), 200 mW ($n=15$), 240 mW($n=13$); (b) Average diffusivity of sodium fluorescein in 80% glycerol solution. The average diffusivities are the average values of the diagonal components of the 3D diffusion tensor. All data depict mean \pm standard deviation.

Reviewers' Comments:

Reviewer #3:

Remarks to the Author:

The changes made to the revised manuscript address the questions raised in the previous submission. The manuscript is recommended for publication without further modification.

Re: NCOMMS-20-36943B

A noninvasive fluorescence imaging-based platform measures 3D anisotropic extracellular diffusion

Response to Reviewer #3:

General Comments:

The changes made to the revised manuscript address the questions raised in the previous submission. The manuscript is recommended for publication without further modification.

Response: We thank the reviewer for their time and their acknowledgement of our efforts in addressing their previous questions.